



# Subdaily meteorological measurements of temperature, direction of the movement of the clouds, and cloud cover in the Late Maunder Minimum by Louis Morin in Paris

Thomas Pliemon[1], Ulrich Foelsche[1,2], Christian Rohr[3,4], and Christian Pfister[3,4]

[1]Institute for Geophysics, Astrophysics and Meteorology/Institute of Physics (IGAM/IP), University of Graz
[2]Wegener Center for Climate and Global Change (WEGC), University of Graz
[3]Oeschger Centre for Climate Change Research, University of Bern
[4]Institute of History, Section of Economic, Social and Environmental History (WSU), University of Bern

**Correspondence:** Thomas Pliemon (thomas.pliemon@uni-graz.at)

**Abstract.** We have digitized three meteorological variables (temperature, direction of the movement of the clouds, and cloud cover) from copies of Louis Morin's original measurements (Source: Institute of History / Oeschger Centre for Climate Change Research, University of Bern) and subjected them to quality analysis to make these data available to the scientific community. Our available data cover the period 1665–1709 (temperature beginning in 1676). We compare the early instrumental tempera-

ture dataset with statistical methods and proxy data to validate the measurements in terms of inhomogeneities and claim that they are, apart from small inhomogeneities, reliable. The Late Maunder Minimum (LMM) is characterized by cold winters and autumns, and moderate springs and summers, with respect to the reference period of 1961–1990. Winter months show a significant lower frequency of westerly direction of movement of the clouds. This reduction of advection from the ocean leads to a cooling in Paris in winter. The influence of the advection becomes apparent when comparing the last decade of the

17[th] century (cold) and the first decade of the 18[th] century (warm). A lower frequency of westerly direction of movement of the clouds can also be seen in summer, but the influence is stronger in winter than in summer. Consequently, the unusually cold winters in the LMM can be attributed to a lower frequency of westerly direction of movement of the clouds. An impact analysis reveals that the winter of 1708/09 was a devastating one with respect of consecutive ice days, although other winters are more pronounced (e.g., the winters of 1676/77, 1678/79, 1683/84, 1692/93, 1694/95 and 1696/97) in terms of mean temperature, ice

days, cold days or consecutive cold days. An investigation of the cloud cover data revealed a high discrepancy in the seasons, where the winter season (DJF) (-13.2 %) and the spring season (MAM) (-12.6 %) show a negative anomaly of the total cloud cover (*TCC*), whereas summer (JJA) (-0.5 %) shows a moderate anomaly of *TCC* with respect to the 30 year mean of the Meteobluedata (1985–2014).

## 1 Introduction

The Little Ice Age (LIA, ca. 1300–1850; e.g., Wanner et al. (2000); Grove (2004); White et al. (2018); Pfister and Wanner (2021)) was one of the coldest periods in the last two millennia (Mann et al., 2009). This is based on evidence from proxy records of both hemispheres (Chambers et al., 2014), which suggest that the LIA was a global phenomenon (Rhodes et al.,





2012). Nonetheless, the term LIA is controversial and it has been suggested to be abandoned because both the temporal duration and the mean magnitude of the cooling of the LIA are far less pronounced than cold periods, which are named ice ages, and
the climate of the LIA was not uniformly cold in space and time (Matthews and Briffa, 2005; Lockwood et al., 2017; Neukom et al., 2019). The Maunder Minimum (MM, 1645–1715) is often regarded as one of the coldest periods of the LIA (Luterbacher et al., 2001; Barriopedro et al., 2008). The coldness reached its climax during the last decades of the MM, which are often called the Late Maunder Minimum (LMM, ca. 1675–1715; Wanner et al. (1995); Slonosky et al. (2001); Legrand and Le Goff (1987)). However, the name Maunder Minimum refers to the study of Walter Maunder „A prolonged sunspot minimum“, who
showed that the sun had little to no sunspots during this time (Lockwood et al., 2017).

Louis Morin lived from 11 July 1635 to 1 March 1715, and he spent his life in Paris (Legrand and Le Goff, 1987, 1992). He achieved the degree Doctor of Medicine and practiced in Paris. In addition to practicing medicine, he measured different meteorological variables for the time-span of 1665–1713 on a daily basis using both early measurement devices and subjective measurements. Therefore, this source can be used to reconstruct the climate during the LMM (Jones, 1999; Pfister, 1999,
2001, 2010). Morin took his measurements in the MM but we will continue with this expression because the majority of his measurements were made in the LMM. The primary focus of earlier studies was on the temperature time series and the pressure time series. For the time period from September 1675 to July 1713, the temperature observations were homogenized and transformed into modern units by Legrand and Le Goff (1987, 1992). The time period from January 1665 to August 1675 was kept untouched, due to the coarse divisions of the thermometer (Legrand and Le Goff, 1987; Legrand et al., 1992; Legrand
and Le Goff, 1992). For our analysis, we digitized the three variables that we give attention to from a copy of Morin's original measurements (Source: Institute of History / Oeschger Centre for Climate Change Research, University of Bern). We include the original data because we want to make them available to the public. Especially for this paper, an analysis of the validity of the data requires looking at the raw data because applying a transfer function (e.g., for temperature) can cause characteristic details of the time series to be lost.

The temperature time series that we obtained was used to analyze the synoptic atmospheric circulation, the variability, and then compare it with auroral activities in the 17[th] century (Legrand et al., 1990, 1992; Diodato et al., 2014). A further correction on the temperature data was made for the time period 1676–1680 (Rousseau, 2013). It was shown that the temperature measurements from 1665 to 1675 are statistically significant on a monthly basis and the period was added to the long term time series of Paris (Rousseau, 2009). Pfister and Bareiss (1994) investigated temperature, wind direction and rainfall observations
for the period 1675–1713 and stated that the temperature data are probably not reliable because in comparison with the Central England Temperature (CET) and Swiss Temperatures there is less cooling over all meteorological seasons. However, because there exist many uncertainties about the details of the measurements, we re-evaluate the temperature series with contemporary temperature measurements, present temperature measurements and proxy data. New, compared to previous studies, is the examination of the time series to possible uncertainties of early instrumental temperature measurements (Camuffo, 2002; Böhm
et al., 2010; Camuffo and Bertolin, 2012; Camuffo and della Valle, 2016; Camuffo et al., 2021), an internal validation (e.g., threshold temperature at 50 % snowfall frequency) and a statistical determination of the measurement time of the observations.





After validation in terms of quality and quantity, we perform an impact analysis to get a clearer picture of the climate in the LMM.

Morin's pressure measurements were homogenized by Cornes (2010); Cornes et al. (2012) on a daily basis, and then used to
create weather maps (Luterbacher et al., 2000; Camuffo et al., 2010) and used for comparison/validation (Slonosky et al., 2001; Können and Brandsma, 2005; Wheeler and Suarez-Dominguez, 2006; Wheeler et al., 2009; Cornes, 2010; Alcoforado et al., 2012). Morin was probably the first individual to observe the dynamics of the free atmosphere (Pfister and Bareiss, 1994). Therefore, we analyzed Morin's observations of the direction of the movement of the clouds and the cloud cover. We extend the time span by 10 years to 1665–1709 for the former and show not only the seasonal mean of the cardinal directions (Pfister
and Bareiss, 1994) but also the interdecadal variability, deviations for the present and indices, which indicate the atmospheric circulation over Europe. By analyzing intraseasonal variability and interannual variability, we show additional evidence (see also Mellado-Cano et al., 2018) that there is a lower frequency of air movement from the west during the winter months and that cold winters have a strong correlation with a low frequency of air movement from the west. We also evaluate the cloud cover, which has not been done before in detail (but was briefly mentioned from Legrand et al. (1992); Legrand and Le Goff
(1992)) , for the period 1665–1709, which allows us to provide a more comprehensive representation of the climate. Due to the lack of contemporary time series of cloud cover, a comparison of the characteristics of cloud cover is only possible with the current time period.





## 2  Data

### 2.1  Metadata and Morin's Meteorological Journal

Not much is known about the details of the measurements that Morin made because did not leave any descriptions, or at least the descriptions are not known so far (Legrand and Le Goff, 1987, 1992). However, important information regarding the metadata can be inferred from his ascetic lifestyle.

#### 2.1.1  The observer Louis Morin and his Meteorological Journal

Louis Morin was born in Le Mans on 11 July 1635 (de Fontenelle, 1715; Lambert, 1751; Delaunay, 1906; Legrand and Le
Goff, 1987, 1992). He received the degree Doctor of Medicine around 1662 and practiced medicine in Paris at the Hôtel-Dieu. On 3 March 1688, he retired to the Abbey of Saint-Victor, which was located on the present site of the Faculty of Sciences of Jussieu. The only document that sheds light on Louis Morin's personality is de Fontenelle's 1715 eulogy to the Académie des Sciences (Legrand and Le Goff, 1987). According to his eulogy, Morin led an ascetic and austere life. From adolescence, he ate bread and water but allowed himself to decorate with a few fruits. Morin died on 1 March 1715, at the age of nearly
eighty, without illness, only for lack of strength. To understand how assiduously and perseveringly Louis Morin made his meteorological observations for nearly half a century, it seems useful to quote the end of Fontenelle's eulogy (translated from de Fontenelle (1715); Legrand and Le Goff (1987)): „This very singular regime was only a part of the daily rule of his life, in which all the functions observed an order almost as uniform and precise as the movements of celestial bodies. He went to bed at seven o'clock in the evening at all times and got up at two o'clock in the morning. He spent three hours in prayer. Between
five and six o'clock in summer, and between six and seven in winter, he went to the Hôtel-Dieu and most often heard Mass at Notre-Dame. On his return he would read the holy scriptures and dine at 11 o'clock. He would then go to the Royal Garden until two o'clock in the afternoon when the weather was fine. There he would examine new plants and thereby satisfying his first and strongest passion. After that he would shut himself up at home, except that he had some poor people to visit, and spend the rest of the day reading books on Medicine, or Erudition, but especially on Medicine because of his duty." An example of
his notes can be seen in Fig. 1 (Legrand and Le Goff, 1987; Pfister and Bareiss, 1994). The first column shows the day of the month. The second column represents the day of lunar cycle. The conjunction, opposition and other aspects of the moon and the sun are given in the third column. The fourth column gives the conjunction, opposition and other aspects of the planets. These last three columns are mostly empty due to the rareness of these events. The thermometer measurements in the fifth column can be divided into three sections, in which Morin used either a different graduation or a different measurement device:
1 February 1665 to 12 April 1670 (nine graduations); 13 April 1670 to 31 August 1675 (11 graduations); 1 September 1675 to 13 July 1713 (140 graduations). The entry consists of a letter (f (du froid) – cold; c (chaud) – warm) and a number. For instance, for the last section (called the big thermometer) the graduation reaches from „f.8.0" (-18.1 °C) to „c.6.0" (35.9 °C). The sixth column shows the hygrometer measurements, which start on 17 May 1701 and continue to the end of his notes. The barometer measurements (seventh column), the direction of the wind (column eight; 16 wind directions; seldom), the
strength of the wind (column nine; 1 (weak) to 4 (strong)), the direction of the movement of the clouds (column 10; 16 wind





directions), the regional origin of air (column 11; 1 to 4), the speed of the clouds (column 12; 1 (slow) to 4 (fast)), the cloud cover (column 13; 0 (sky free) to 4 (cloudy)), the intensity and duration of rainfall (column 14; Intensity (first number) and duration (second number)). These variables were measured throughout the period with just a few gaps, except for column eight. The impressive consequence of his observations can be seen in Tab. 1, in which the quality of the variables of our interest are

shown in percentage of completeness. Column 15 shows fog (b.intensity), snow (n.intensity.duration), small hailstones, thunder perihelion and colour of sky and the sixteenth column miscellaneous observations, such as earthquakes, comets and halos.

|  | Morning / % | Midday / % | Evening / % | Per Day / % |
|---|---|---|---|---|
| Temperature | 99.1 | 98.4 | 98.6 | 99.6 |
| Direction of the movement of the clouds | 70.0 | 80.4 | 64.5 | 89.8 |
| Cloud Cover | 96.7 | 95.4 | 94.6 | 98.2 |

**Table 1.** Quality in terms of missing data, distinct between the three measurements in the morning, on midday/noon and in the evening. The last column shows the percentage of days on which at least one measurement was taken.





**Figure 1.** Example of Morin's notes (Source: Institute of History / Oeschger Centre for Climate Change Research, University of Bern). Column one shows the day of the month; column two shows the day of lunar cycle; column three shows the conjunction, opposition and other aspects of the moon and the sun; column four shows the conjunction, opposition and other aspects of the planets; column five shows the thermometer measurements; column six shows the hygrometer measurements; column seven shows the barometer measurements; column eight shows the wind direction; column nine shows the wind strength; column 10 shows the direction of the movement of the clouds; column 11 regional origin of air; column 12 shows the speed of the clouds; column 13 shows the cloud cover; column 14 shows intensity and duration of rainfall; column 15 shows fog, snow, small hailstones, thunder perihelion and colour of sky; column 16 gives miscellaneous observations, such as earthquakes, comets and halos. For more details, see the text.





### 2.1.2 Time of the observations

Most of the time, Morin made three observations per day, but there are at least some days on which he measured four times. The specific measurement times are uncertain but due to his ascetic life it is suggested that he did the first measurement around 6 am, the second measurement between 11 am and 2 pm, and the third measurement between 6 and 7 pm (Pfister and Bareiss, 1994). Cornes et al. (2012) estimated the observation times: 6 am, 3 pm, and 7 pm.

### 2.1.3 Location of the observations

The location of the so called big thermometer changed several times, which is known from the rare notes of Louis Morin (Legrand and Le Goff, 1987). Until October 1685, he lived in the Quinquempoix Street (Number 1 in Fig. 2), then until June 1688 in the Hotel Rohan-Soubisse (Number 2 in Fig. 2), where the National Archives are located today. Until his death in 1715, he lived in the abbey Saint-Victor (Number 3 in Fig. 2), which is located at the city border next to the Seine. (Legrand and Le Goff, 1987; Pfister and Bareiss, 1994)

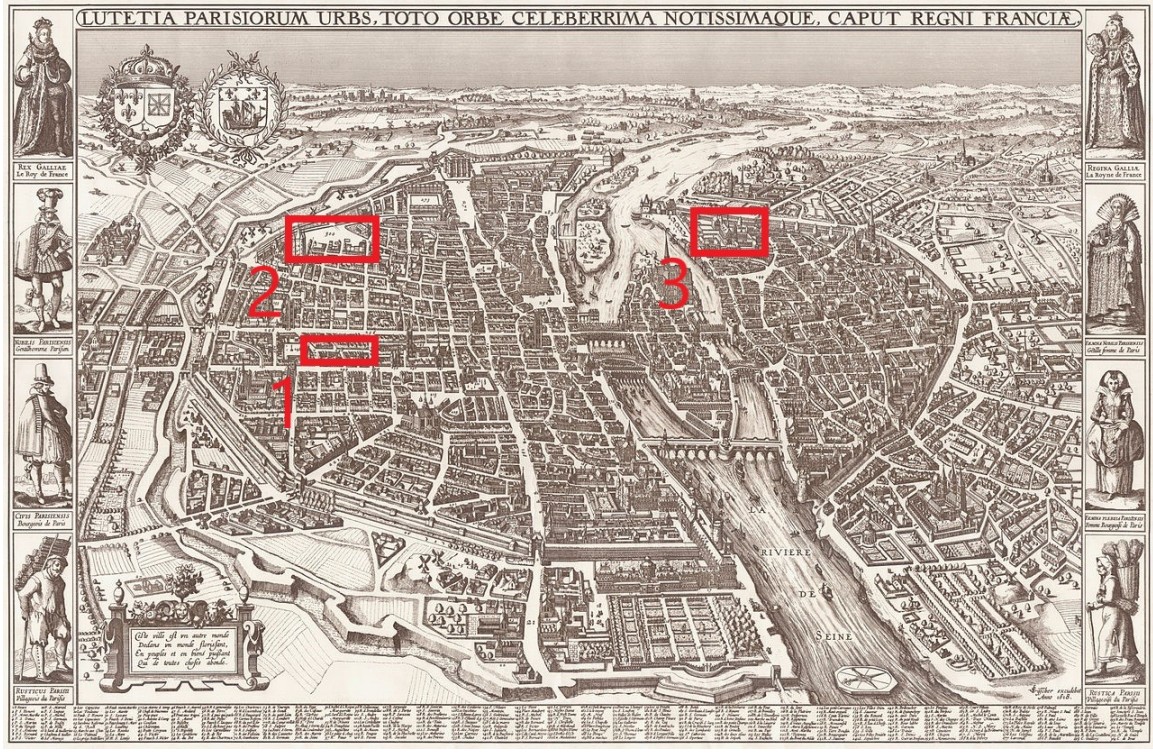

**Figure 2.** A map of Paris (Visscher, 1618), the marked locations show where Morin lived. Until October 1685, he lived in the Quinquempoix Street (1); then until June 1688 in the Hotel Rohan-Soubisse (2), where the National Archives are located today; and until his death in 1715 he lived in the abbey Saint-Victor (3), which is located at the city border next to the Seine.



## 2.2 Reference data

A simulated historical weather dataset developed by Meteoblue.com (Meteoblue, 2020) was used to compare the direction
of the movements of the clouds and the cloudiness to contemporary conditions (1985–2014). As reference period for the
temperature data, we used the observations of E-OBS (Cornes et al., 2018) for the common period from 1961–1990. For both,
we chose a reference period of 30 years as suggested by the WMO (WMO, 2003). Furthermore, we included in our study the
Grape Harvest Dates (GHD) from Beaune (Labbé et al., 2019), the Central England Temperature (CET, Manley (1974)) and
winter temperatures from De Bilt (van der Meulen and Brandsma, 2008; Brandsma and van der Meulen, 2008).

## 130 3 Methods

### 3.1 Digitalization process

Although we did get digitized data from Christian Rohr and Christian Pfister, given that it had already been processed and
did not cover the entire time span of the copies, we completed the data digitization for the years of 1665 to 1709. Thus, we
digitalized 137926 values and checked the digitized data due to implausible values and typos. For temperature data, we further
consulted already digitized data from Bern to check for mistakes.

### 3.2 Missing data

We kept the original data and did not correct the missing values, except when performing the impact analysis for the temperature. In detail, smaller gaps with one or two consecutive missing values were filled by the mean of the day before and the
day after the missing value. Thus, we filled up 60 values for the morning temperature, 132 values for the midday/afternoon
temperature, and 113 values for the evening temperature.

### 3.3 Calibration and homogenization of the thermometer

We know that the Florentine Thermometer was already known in France (Paris) because it is documented that Ismaël Boulliaud
got one in 1658 (Legrand and Le Goff, 1987; Rousseau, 2013; Camuffo et al., 2020). The conventional graduation of the
Florentine Thermometer is different to the one used by Morin. It is supposed that he used three different thermometers because
the graduations of the temperature measurements differ viewed over time. The first from 1 February 1665 to 12 April 1670
with nine divisions. The second from 13 April 1670 to 31 August 1675 with 11 divisions. The third (big thermometer) from
1 October 1675 to 13 July 1713 with 15 divisions. Furthermore, each of these divisions are separated in 10 parts (See Legrand
and Le Goff (1987) Fig. 2). The original notes from 1710 to 1713 were not available to us. It is supposed that Morin used
the following calibration: put the thermometer in ice, in which salt was added. After an appropriate time, while keeping the
150 thermometer in the ice, mark the position of the spirit. Then put the thermometer into a cellar, which is cut off from the outside
world. Measure the temperature again and make a mark. Split the space into 15 divisions and label the marks with numbers,
using the mark in the cellar as point of origin (Legrand and Le Goff, 1987; Rousseau, 2009). Due to the low resolution of





the thermometers used before 1675, we only used the big thermometer for further calculations. Legrand and Le Goff (1987) showed a linear relationship between the maximum temperatures, the minimum temperatures and the measured temperature 155 (TM) of the two time periods and used three different linear regression formulas for the maximum temperatures (1), for general minimum temperatures (2) and for significant cold minimum temperatures (3):

$$Tmax = 14.3 + 3.60TM \tag{1}$$

$$TM > f.5,5 : Tmin = 12.2 + 3.33TM \tag{2}$$

160

$$TM < f.5,5 : Tmin = 18.9 + 4.63TM \tag{3}$$

The entry of TM consists of a letter (f (du froid) – cold; c (chaud) – warm) and a number (See Fig. 1). For entries which are noted with c, TM is multiplied with the following value; and for entries which are noted with f, TM is multiplied with the negative value of the following value. We used this calibration to receive values in degree Celsius. Furthermore Rousseau 165 (2013) suggested that the years from 1676 to 1680 should be corrected because harvest dates differed in that period significantly from the temperature measurements of Morin. So, Rousseau (2013) compared the monthly means of the Morin measurements with the CET and modified the months between 1676 and 1680 with the following values (Jan. to Dec.): {-1.0, -1.5, -1.5, -1.5, -1.5, -1.0, -1.0, -1.3, -1.8, -1.0, 0.0, -0.5}, because he found significant differences compared to the CET. However, we did not apply this proposed change because a comparison with grape harvest dates (*GHD*), snow frequency, and contemporaneous 170 measurement periods show deviations (CET (Manley, 1974) and De Bilt winter temperatures (van der Meulen and Brandsma, 2008; Brandsma and van der Meulen, 2008)), but the magnitude of the change seems to us to be too large. Furthermore, because this change refers to the daily mean, it is difficult to make an exact attribution to the minimum and maximum temperature. The extension to months outside the growing season is also debatable. Another point is that there is no documented change in his circumstances in terms of location change and so on in this period. However, it has to be mentioned that we do not disagree 175 that the temperatures are unusually high for these years noted by Morin (see Tab. 2).

### 3.4 Regression analysis

When a regression curve was used to visualize the data, we used the prebuilt geom_smooth() function of R. It performs a locally weighted (tricube weighting) polynomial (degree 2) regression. We defined the parameter span = 0.25, which means that 25 % of the total data length per data point is considered for the regression.





## 4 Results

### 4.1 Temperature

Based on the daily available temperature data, we perform an impact analysis. However, to check the reliability of the values, it is necessary to perform a statistical analysis beforehand and to compare them with proxy data.

#### 4.1.1 Qualitative and quantitative cross-validation

As previously mentioned, neither the measurement times, the measuring device used, nor the location of the measuring device are explicitly noted by Morin and are therefore based on assumptions. Consequently, it is necessary to validate Morin's measurement data with alternative methods to conclude to metadata. Although cross-validations with other measurement series are possible, only the CET measurement series exists at the same time. This series can only be used as a limited comparison because the climate in Great Britain differs from that on the mainland. This leaves only statistical possibilities and the use of proxy data.

Due to the fact that the measurement times can only be determined based on Morin's ascetic lifestyle, we looked for a way to validate these measurement times. Given that the average of the diurnal variation of the temperature follows a unimodal distribution, we performed a statistical approach by calculating the number of days on which the temperature, which usually indicates a lower temperature value, exceeds the usually higher temperature (e.g., $T_{mi} < T_{ev}$). After that, we compared the results with the Meteobluedata for the reference period from 1986–2015. In detail, we calculated the time index (*TI*) after the following formula:

$$TI = |\frac{n_{T_{mi} < T_{ev}} - n_{T_i < T_j}(Ref)}{n_{T_{mi} < T_{ev}}}| + |\frac{n_{T_{mo} < T_{ev}} - n_{T_k < T_j}(Ref)}{n_{T_{mo} < T_{ev}}}| + |\frac{n_{T_{mo} < T_{mi}} - n_{T_k < T_i}(Ref)}{n_{T_{mo} < T_{mi}}}|$$

Thereby, the measurements of Morin are indicated as morning temperature ($T_{mo}$), midday temperature ($T_{mi}$) and evening temperature ($T_{ev}$). Because earlier papers are based on suggestions, when the measurements of Morin were done most properly, we integrated the Meteobluedata with the variables $T_i$, $T_j$ and $T_k$, where $i \in \{4-5,...,7-8\}$, $j \in \{13-14,...,16-17\}$ and $k \in \{17-18,...,20-21\}$. So, $n_{T_{17-18} < T_{14-15}(Ref)}$ means for example the number of days, when the mean temperature of 17:00 and 18:00 exceeds the mean temperature of 14:00 and 15:00. We calculated the *TI* multiple times for reasonable time triples $\{T_i, T_j, T_k\}$ (e.g., $\{T_{6-7}, T_{14-15}, T_{18-19}\}$ and looked for the minimum of *TI*, which can be seen as the sum of the normalized linear anomaly of each measurement. This method can only be used as an indicator because there are many underlying premises that have to be committed. Nonetheless, we think that by taking a longer time period, this method can be seen as an additional indicator for making a statement about Morin's measurement times. The results showed a minimum of $TI = 0.92$ for the time triples $\{T_{5-6}, T_{15-16}, T_{18-19}\}$ and $\{T_{6-7}, T_{15-16}, T_{18-19}\}$. On the one hand, this supports the hypothesis of the measurement times of earlier papers (Legrand and Le Goff, 1987; Cornes et al., 2012). On the other hand, it is a practical advantage for the analysis of the temperature measurements because Morin performed a measurement near the daily minimum and maximum. Nevertheless, taking a closer look at the annual distribution of the number of days when $T_{ev} > T_{mi}$, it can be seen that there is a strong variation, whereas there is a nearly uniform distribution on a monthly basis (see Fig. A1).





Morin also documented snow days and rainy days. This allows us to determine the temperature at which snowfall occurs, and vice versa. It is generally assumed that snowflakes begin to melt at a temperature of 0 °C. However, this is not true in detail because there are spatial differences because the melting point is dependent on pressure and even more on humidity. Increasing pressure leads to an increasing melting point temperature and increasing humidity leads also to an increasing melting point temperature. It is found that the phase transition occurs over a fairly wide temperature range, from about -2 °C to

+4 °C, over (low-elevation) land and from -3 °C to +6 °C over ocean (Dai, 2008). Jennings et al. (2018) found that the air temperature at which rain and snow fall in equal frequency varies significantly across the Northern Hemisphere, averaging 1.0 °C and ranging from -0.4 to 2.4 °C for 95 % confidence interval. The results of Jennings et al. (2018) showed that the threshold temperature is about 1.5 °C in Paris. This means that an investigation of the melting point temperature or snowfall frequency can provide information about whether it is an indoor or outdoor measurement, or if the lower calibration point

is accurate. The case where deviations cancel themselves is unlikely. So, if the snowfall frequency, calculated from Morin's measurements, is somewhat equal to 1.5 °C, then we assume that he performed an outdoor measurement and that the lower calibration point is accurate. The results are shown in Fig. 3, the solid black line shows the snowfall frequency as a function of the temperature for all measurements. The green, orange, and blue lines represent the results for the measurements in the morning, on midday/afternoon, and in the evening, respectively. Because the snowfall frequency is dependent on pressure and

humidity, it is not anomalous and there is a significant difference between the measurements on different times of the day. The red line shows the data of the Meteoblue dataset. The threshold temperature at 50 % of the Meteoblue dataset equals 1.75 °C and 1.5 °C for the Morin dataset. Even though the threshold temperature of the snowfall frequency at 50 % fits well, the slope of the function is smaller than expected. The reason for this can be found in Morin's records because he noted three measurements per day for the temperature, whereas he varied between three (majority) and six notes per day for snow and rain

measurements. This means that a distinct attribution between temperature and snow/rain cannot always be done, which could lead to a function with a smaller slope.

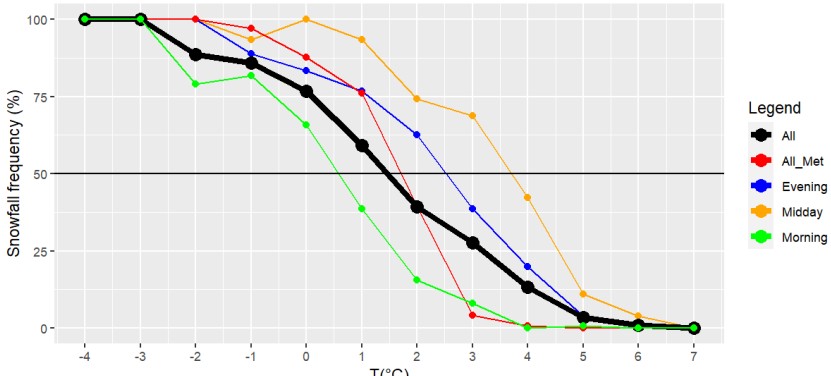

**Figure 3.** Snowfall frequency curves calculated using Morin's observations of the morning (green), midday/afternoon (orange), and the evening (blue). The black line includes all of Morin's measurements and the red line shows the mean of the Meteobluedata.



Camuffo and della Valle (2016); Camuffo et al. (2017) suggested an index to recognize indoor/outdoor exposures and room ventilation. This is a useful index because it can point out the buffering capacity of the building structure on an hourly time scale, which is named the normalized diurnal range (*NDR*) (i.e., the ratio between the observed diurnal temperature range, or a given portion of it, and the same but measured outdoors in the 1961–1990 reference period). The diurnal range has a marked seasonal character. Normalization is applied to remove the seasonal dependence and to point out how much the observations are damped in comparison with the outdoor cycle. *NDR* can be considered to be a room fingerprint, which is strictly linked to the room ventilation, solar exposure and use. In this paper, *NDR* has been calculated with the formula

$$NDR = \frac{\sum_n^{n_{SP}} \frac{T_{n,15}-T_{n,6}}{2n_{SP}}}{\sum_n^{n_{RP}} \frac{T_{n,15}-T_{n,6}}{2n_{RP}}}$$

where $T_{n,6}$ and $T_{n,15}$ are the readings of the nth calendar day at 06:00 and 15:00; $T_{n,6}$ is close to the daily minimum and $T_{n,15}$ to the daily maximum and the difference $T_{n,15}$ - $T_{n,6}$ equals the diurnal range of the nth calendar day; $n_{SP}$ is the total number of days of the selected period (SP); the label RP refers to the 1961–1990 reference period (E-OBS). $NDR = 0$ when the buffering capacity is very large and the ventilation is poor; $NDR = 1$ when the thermometer is kept outside, well ventilated and far from walls such as modern weather stations. The formula calculated with our datasets leads to a value of 0.90. So we assume that the temperature measurement was performed outside and it was well ventilated with small influences from walls and so on. However, if one divides the periods into the three different localities where Morin lived, then differences arise for this parameter. We calculate $NDR = 0.80$ from 1675 to October 1685, $NDR = 0.80$ from November 1685 to June 1688, and $NDR = 0.95$ from July 1688 to 1709. Analog to that, these differences can be seen when calculating the diurnal temperature range (*DTR*; see Fig. A2).

Another possibility to validate the temperature data is to compare it with proxy data, such as with grape harvest dates (*GHD*). Due to the high control of temperature on grape ripening, *GHD* are linked to the growing season temperature (*GST*: mean (e.g., Chuine et al. (2004); Meier et al. (2007); Labbé et al. (2019) ) or maximum temperature (e.g., Daux et al. (2012)) averaged over April to July (e.g., Labbé et al. (2019)), April to August (e.g., Chuine et al. (2004); Meier et al. (2007)) or April to September (e.g., Daux et al. (2012)). Several authors have evidenced a statistical significant correlation between *GST* and *GHD*. For example, Daux et al. (2012) showed that high correlations are obtained between GHD regional composite series and the temperature series of the nearest weather stations. Moreover, the *GHD* regional composite series/temperature correlation maps show spatial patterns similar to temperature correlation maps (Daux et al., 2012). Furthermore, others have quantified the change of *GHD* in terms of the variation of the temperature. Menzel (2005) stated that the variation of the *GHD* of a mean European series was 10.0±0.9 days for 1 °C variation of the *GST*. Meier et al. (2007) showed that 12 days of grape harvest difference correspond to around 1 °C April to August temperature for varieties grown in Switzerland. Chuine et al. (2004) calculated a linear correlation coefficient between reconstructed temperatures and *GHD* of 0.75 ± 0.07 for Paris. However, Labbé et al. (2019) constructed *GST* in a different way with a statistical model, which led to a rate of 10 days / 0.61 °C. The difference of his calculations is that he weighted the months differently and used the months from April to July.

In Fig. 4, we have plotted the mean temperature of the months April to August for comparison to the *GHD*. The calculated correlation between *GHD* and $T_m$ is -0.76 (-0.50 for $T_{max}$). Compared to the present time, the *GHD* of the LMM





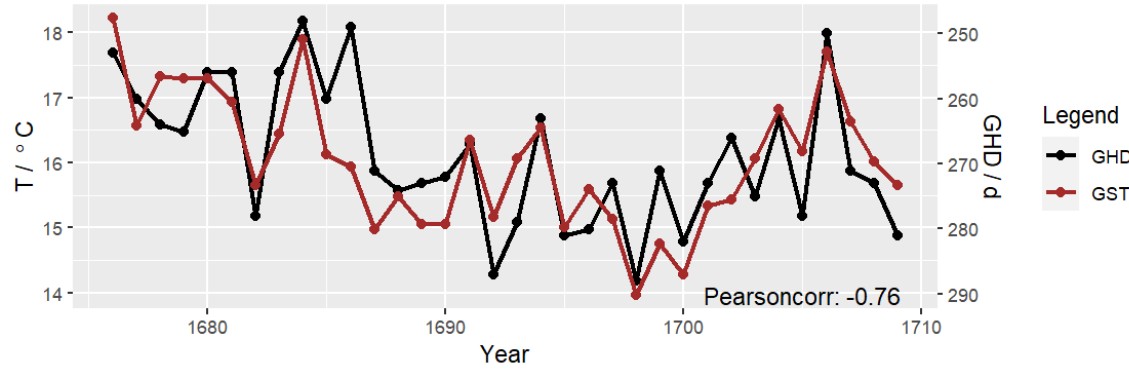

H

**Figure 4.** Comparison between *GST* of $T_m$ in Paris and the grape harvest dates (*GHD*) from Dijon (Labbé et al., 2019) in the LMM. The Pearson correlation index equals to a value of -0.76.

was slightly earlier than in the reference period from 1961–1990. Surprisingly the mean temperature of the months April to August is higher in the LMM than the mean temperature of the reference period from 1961–1990. So, we obtain over

the time periods: $GHD_{Morin} = 269.85 \pm 10.83 (GHD_{1961-1990} = 270.20 \pm 7.96)$. This means, that the temperature difference should be more or less 0 between these time periods, which is fulfilled for the mean of the maximum temperature: $GST_{Morin} = 20.47 \pm 1.29 (GST_{1961-1990} = 20.23 \pm 1.07)$. A slightly higher difference occurs when looking at the mean temperature: $GST_{Morin} = 16.00 \pm 1.02 (GST_{1961-1990} = 15.45 \pm 0.75)$.

We examined Morin's measurements for inhomogeneities with respect to a non-northward orientation of the thermometer

(Böhm et al., 2010) and inhomogeneities caused by the use of alcohol thermometers (Camuffo and della Valle, 2016). No abnormalities seem to occur on either point.

### 4.1.2 Seasonal temperature anomalies

The LMM is generally considered to be the second coldest period in the LIA, after the Spörer Minimum. Although it is a short period, the characteristics are different and interdecadal variability is high. Fig. 5 shows the seasonal temperature anomalies of

each meteorological season in reference to the mean value of 1961–1990. The solid lines represent the regression curves for each meteorological season. The anomalies show that the general statement that the LMM was a cold period can be discussed; that is, warmer months (MAM, JJA) reflect small anomalies and colder months (SON, DJF) show negative deviations with respect to the reference period. A strong variability can be observed in the DJF season, which causes extremes for $T_{min}$ and $T_{max}$ between 1675 and 1690, almost year after year. The decade from 1691 to 1700 is known as the coldest period in the

LMM, which is confirmed in the Fig. 5 for each season, stronger for $T_{max}$ (-1.9 °C anomaly), somewhat weaker for $T_{min}$ (-1.8 °C anomaly). The decade from 1701 to 1710 experiences warming across all seasons, strongly pronounced in JJA and DJF, somewhat weaker in MAM and SON. An interesting aspect is that $T_{max}$ between 1691 and 1700 shows several years with positive anomalies for MAM and JJA. This also can be seen in the *GHD*, which do not reflect the cold period in the LMM.



In summary, the decades 1691–1700 and 1701–1710 stand in contrast: the former as a pronounced cold period and the latter as a warm period. Tab. 2 shows the anomalies of $T_{min}$, $T_{max}$ and $T_{mean} = 0.5 T_{min} + 0.5 T_{max}$ separated into decades and seasons. Extraordinary positive anomalies appear between 1675–1680 for MAM and JJA (cf. section 3.3) and extraordinary negative anomalies appear as expected in DJF and SON in the last two decades of the 17[th] century. With respect to the mean temperature three DJF seasons fall below an anomaly of -4 °C, namely the winter of 1683/84, 1694/95 and 1696/97. Interestingly, the notorious winter of 1708/09 is not particularly pronounced in terms of mean temperature because December 1708 was moderate and devastating cold temperatures ($T_{min}$ up to -18 °C) were restricted to January.

| | 1675–1680 / °C | 1681–1690 / °C | 1691–1700 / °C | 1701–1709 / °C | 1675–1709 / °C |
|---|---|---|---|---|---|
| $T_{min_{DJF}}$ | 0.3 | -0.4 | -1.8 | 1.0 | -0.3 |
| $T_{min_{MAM}}$ | 2.0 | 1.1 | -0.4 | 0.1 | 0.5 |
| $T_{min_{JJA}}$ | 2.3 | 1.0 | -0.1 | -0.2 | 0.5 |
| $T_{min_{SON}}$ | 0.8 | 0.8 | -1.1 | -0.8 | -0.2 |
| $T_{mean_{DJF}}(T_{CET_{DJF}})$ | -0.3 (-1.3) | -1.0 (-0.9) | -1.8 (-1.6) | 0.2 (-0.6) | -0.8 (-1.0) |
| $T_{mean_{MAM}}(T_{CET_{MAM}})$ | 2.0 (-0.7) | 0.4 (-0.4) | -0.2 (-1.4) | 0.7 (-0.2) | 0.5 (-0.7) |
| $T_{mean_{JJA}}(T_{CET_{JJA}})$ | 1.7 (0.0) | 0.1 (-0.5) | -0.3 (-0.9) | 0.7 (0.2) | 0.5 (-0.3) |
| $T_{mean_{SON}}(T_{CET_{SON}})$ | 0.1 (-1.0) | -0.3 (-1.1) | -1.5 (-1.7) | -0.7 (-0.7) | -0.7 (-1.1) |
| $T_{max_{DJF}}$ | -0.9 | -1.6 | -1.9 | 0.2 | -1.1 |
| $T_{max_{MAM}}$ | 1.9 | -0.3 | 0.0 | 1.3 | 0.5 |
| $T_{max_{JJA}}$ | 1.1 | -0.8 | -0.5 | 1.4 | 0.2 |
| $T_{max_{SON}}$ | -0.5 | -1.3 | -1.8 | -0.6 | -1.1 |

**Table 2.** Anomalies of $T_{min}$, $T_{max}$ and $T_{mean}$ divided into seasons (DJF, MAM, JJA and SON) with respect to the reference period (E-OBS: 1961–1990). The CET is given as a comparison in the brackets of the mean temperatures.

### 4.1.3 Impact analysis

To be able to examine the different characteristics of this period in more detail, we have carried out an impact analysis. An impact analysis shows a more detailed picture of the circumstances or the influence of the weather/climate on humans. Consecutive warm and cold days dramatically complicate living conditions and lead to lower crop yields, famine, and so on. For the cold season (DJF), the indices *FD0* (number of days on which the temperature falls below 0 °C) and *ID0* (number of days on which the temperature stays below 0 °C) were calculated. The results (See Fig. 6) reveal a high variation of cold winters, which occur more frequently in the decade 1690–1700. However, some devastating winters with a high number of ice days (*ID0* 20–30) are found in the last decade of the 17[th] century , namely: 1676/77, 1678/79, 1680/81, 1684/85 and 1708/09. Only the winters 1694/1695 and 1696/97 are found in the cold decade. However, when measured by the number of frost days, the last decade of the 17[th] century remains the coldest period.





**Figure 5.** Seasonal temperature anomalies (DJF: December, January, February; MAM: March, April, May; JJA: June, July, August; SON: September, October, November) for (a) minimum, (b) maximum, and (c) mean temperature with respect to the mean value from 1961–1990 (E-OBS). The indicators for each year and the regression curves are shown. In addition, the annual mean temperature is shown in (c).





For the warm season (JJA), the indices *SU25* (days with a maximum temperature value above 25 °C / summer days) and *SU30* (days with a temperature value above 30 °C / heat days) were examined. Up to the year 1690, there is a strong variability with alternating higher and lower values for *SU25*, as well as a gradually disappearing *SU30* value. This follows a decade with moderate summers in the decade 1691–1700 (i.e., moderate *SU25* values and *SU30* values towards zero). The warmer first

decade of the 18th century is also evident here with high *SU25* values (>40 days for the summers of 1701, 1704, 1705, 1706, and 1707) and also with an increase in *SU30* values in the same years. This again confirms that the summer months in the LMM can be described as moderate.

|  | 1675–1680 / d | 1681–1690 / d | 1691–1700 / d | 1701–1709 / d | 1675–1709 / d |
|---|---|---|---|---|---|
| *FD0* | -2.2 | 0.7 | 12.8 | -8.7 | 1.3 |
| *ID0* | 4.0 | 4.3 | 6.5 | -2.0 | 3.2 |
| *SU25* | 8.9 | -7.0 | -7.8 | 12.1 | 0.2 |
| *SU30* | -1.2 | -2.6 | -4.1 | 0.3 | -2.0 |
| *Cons. cold days* | 4.5 | 2.1 | 4.7 | -2.0 | 2.1 |
| *Cons. ice days* | 1.5 | 2.1 | 1.0 | 0.0 | 1.1 |
| *Warmwave* | 1.2 | -2.5 | -2.4 | 1.8 | -0.8 |
| *Heatwave* | 0.0 | -1.3 | -2.4 | -0.6 | -1.2 |

**Table 3.** Anomalies of *FD0, ID0, Cons. cold days* and *cons. ice days* for the season DJF and *SU25, SU30, warmwaves* and *heatwaves* for the season JJA with respect to the reference period (E-OBS: 1961 1990).

An even more differentiated picture emerges when looking at the consecutive cold days, or ice days (Fig. 7). Winter 1676/77 (42 consecutive cold days) stands out clearly. On average, the last period of the 17th century is again the most striking, but

compared to the winter 1676/77 these appear moderate with respect to consecutive cold days. The winter of 1708/09, which is known as a devastating winter, has the most ice days. This winter is characterized of a mild December, a freezing cold January with temperatures down to -18 °C and a cold February. This is why the winter of 1708/09 is not outstanding with respect to the mean of $T_{min}$, $T_{max}$ and *FD0*. The most consecutive warm days and heat days are again to be found in the first decade of the 18th century. In summary, the picture is more differentiated and exceptional weather conditions cannot be limited to

the supposedly cold period of the last decade of the 17th century and the supposedly warm period of the first decade of the 18th century.





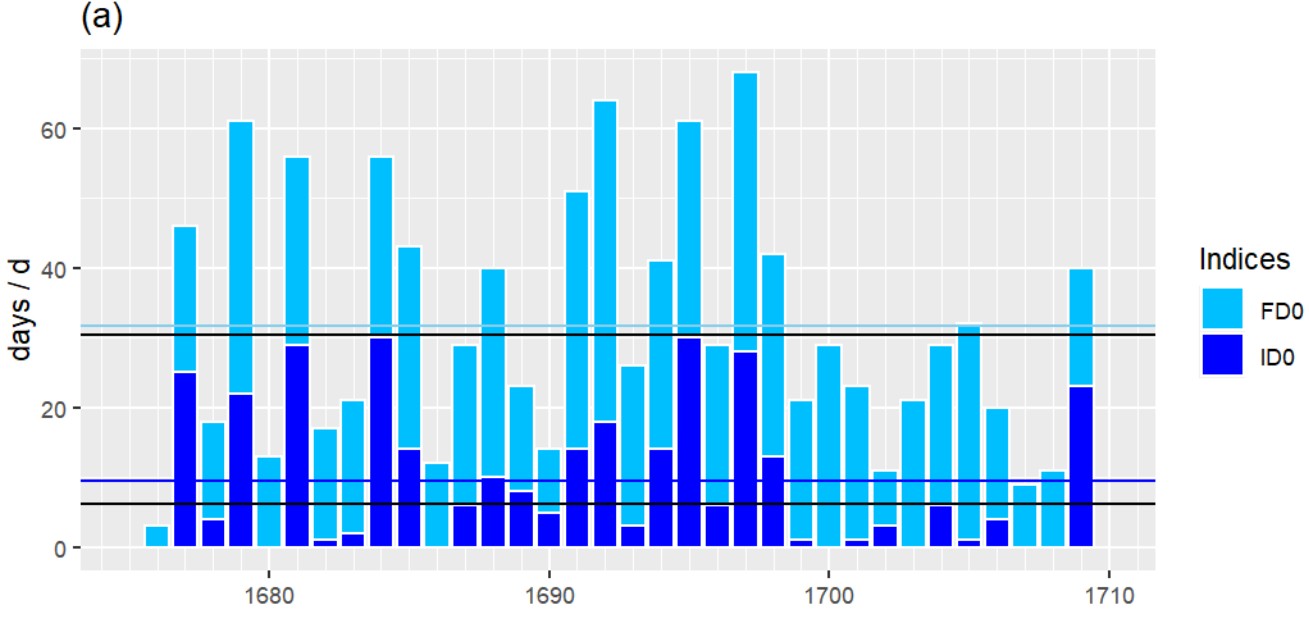

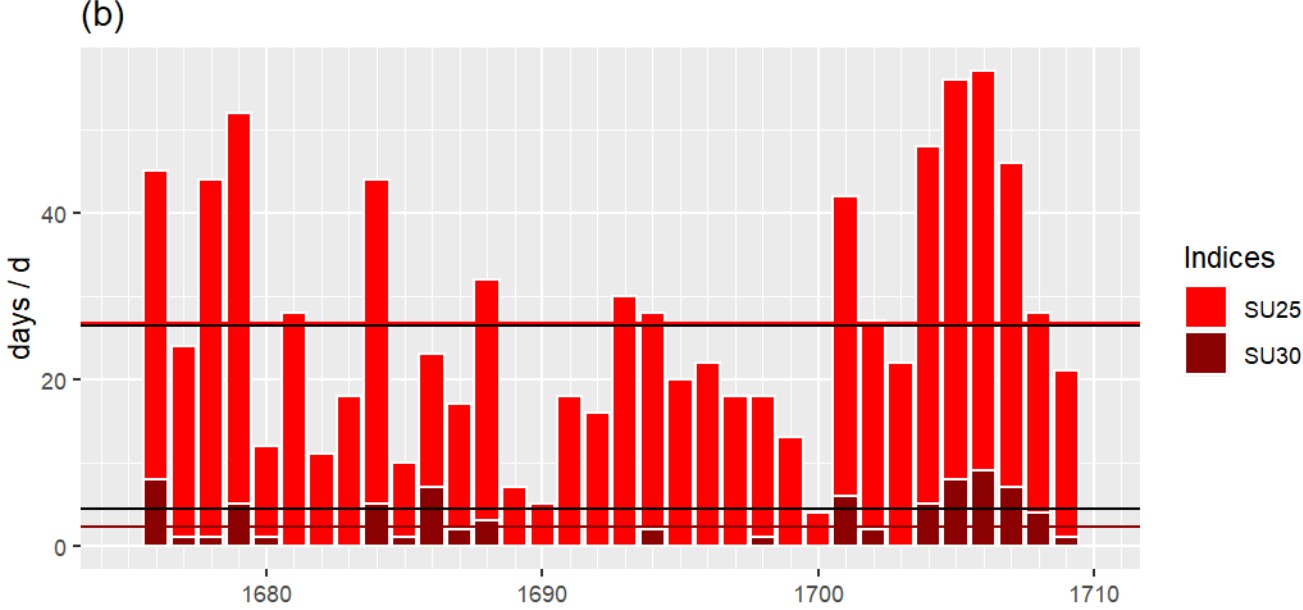

**Figure 6.** (a) Calculation of the indices *FD0* (number of days on which the temperature falls below 0 °C) and *ID0* (number of days on which the temperature stays below 0 °C) for the cold season (DJF). The horizontal lines indicate the respective average values over the entire period ($\overline{FD0}$: skyblue; $\overline{ID0}$: blue; $\overline{FD0}_{Ref}$: upper black line; $\overline{ID0}_{Ref}$: lower black line). Meanwhile, (b) shows the calculation of the indices *SU25* (days with a temperature value above 25 °C) and *SU30* (days with a temperature value above 30 °C) for the warm season (JJA). The horizontal lines indicate the respective average values over the entire period ($\overline{SU25}$: red; $\overline{SU30}$: dark red; $\overline{SU25}_{Ref}$: upper black line; $\overline{SU30}_{Ref}$: lower black line).



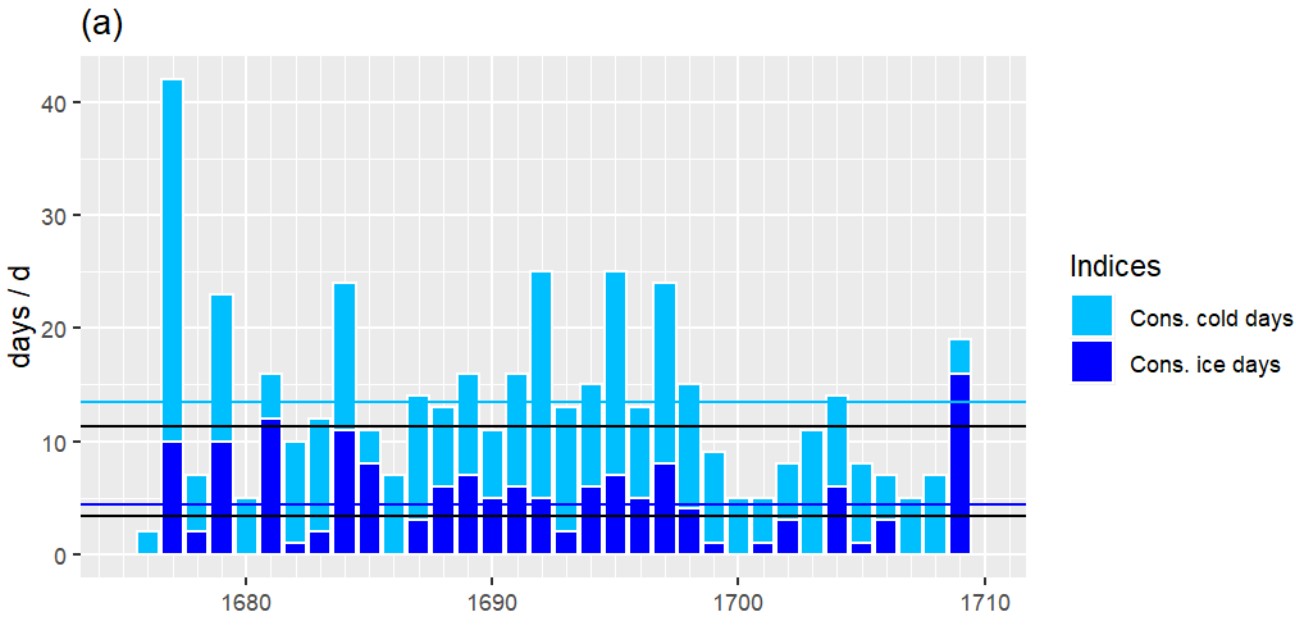

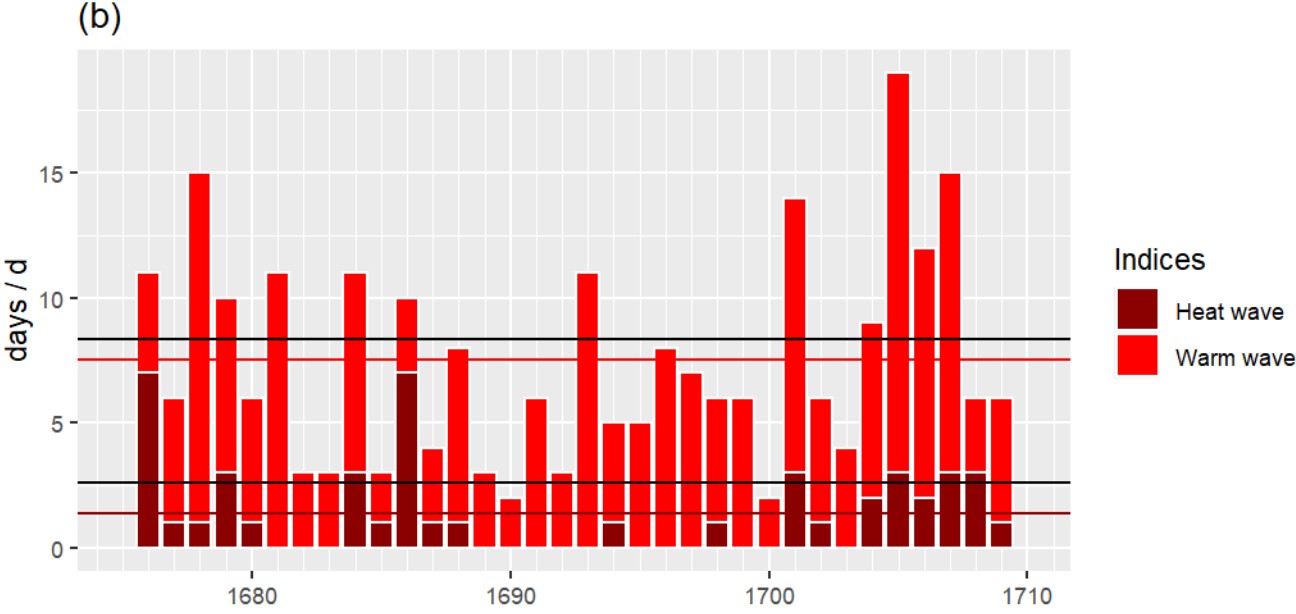

**Figure 7.** Calculation of (a) maximum number of consecutive cold days and maximum number of consecutive ice days for the cold season (DJF). The horizontal skyblue line indicates the mean of consecutive cold days and the blue line the mean of consecutive ice days. The horizontal upper (lower) black line indicates the mean of consecutive cold (ice) days of the reference period. Meanwhile, (b) shows warm waves (Consecutive days of Tmax > 25 °C) and heat waves (consecutive days of Tmax > 30 °C) for the summer season (JJA). The horizontal red line indicates the mean of warmwaves and the dark red line the mean of heatwaves. The horizontal upper (lower) black line indicates the mean of warmwaves (heatwaves) of the reference period.





## 4.2 Cloud Cover

The scale of Morin's readings reaches from 0 to 4, where 0 means sky completely free and 4 means sky completely cloudy (analog to his notes of fog). The readings can be assumed to represent the total cloud cover (*TCC*). *TCC* series were frequently
given in oktas (WMO (2021); Fine: 0 (Sky clear) to 2 (2/8 of sky covered) oktas; Partly cloudy: 3 to 5 oktas; Cloudy: 6 to 7 oktas; Overcast: 8 oktas (Sky completely covered); Sky obstructed from view: 9 oktas (e.g., fog). To obtain comparable quantities, Morin's original data are multiplied with the factor 2. Because Morin also recorded foggy days, we weighted records that had values of 3 and 4 by 9 oktas. This allows a quantitative comparison to present *TCC* series. In Fig. 8 the *TCC* series is plotted from 1665–1709. The yearly mean is shown at the bottom and above the *TCC* series is separated into the seasons DJF,
MAM, JJA and SON. A remarkably high *TCC* appeared in the years 1692 and 1693. Whereas, the highest value seasonally appeared in the year 1668. Remarkably low values appeared in autumn 1669 and in winter 1691. A linear regression analysis shows an increase in *TCC* over the investigated time period (Fig. 8). The slope of the linear regression accounts for 0.007 in DJF, 0.013 in MAM, 0.012 in JJA, and 0.012 in SON per year. Clouds contribute both to cooling and warming the Earth's surface, due to their high albedo and capacity to absorb infrared radiation (Mace et al., 2006; Zelinka and Hartmann, 2010).
Consequently, a statement about a correlation of temperature and cloud cover is difficult. Furthermore, traditional observations lack objectivity; for example, regarding the estimation of *TCC* and the determination of cloud types (Sanchez-Lorenzo et al., 2012). In this context, changes of observing practices, time of observations, and different observers can introduce biases that affect the homogeneity of the series (Sanchez-Lorenzo et al., 2012). Due to these reasons, a quantitative comparison between Morin's *TCC* series and the present *TCC* series is problematic. However, several publications present the relationships between
observed world-wide long-term decrease in daily temperature range (*DTR*) and the systematic increase in cloud cover (Dai et al., 1999; Sun et al., 2000; Cox et al., 2020), and trends can be found in different seasons and different decades. In Tab. 4 the percentage change of Morin's *TCC* series is shown per decade, with the mean of the meteoblue dataset from 1986 to 2015 as reference. The seasonally mean *TCC* from 1665–1709 shows a significant lower value for seasons DJF, MAM and SON, while the magnitude of *TCC* stays roughly the same in JJA. Only the season JJA obtains higher values, respectively, in the decades
from 1681 to 1709. The highest anomalies are observed in 1665–1680, when the spring season has a 23.1 % (1665–1670) and a 22.5 % (1671–1680) lower value with respect to the mean of the Meteoblue dataset (1986–2015). Over the entire series from 1665–1709 the highest anomalies, with respect to the mean of present *TCC* series, are obtained in DJF (-13.2 %) and MAM (-12.6 %). A moderate anomaly can be seen in SON (-7.7 %) and a vanishing anomaly is obtained in JJA (-0.5 %).

|      | 1665–1670 / % | 1671–1680 / % | 1681–1690 / % | 1691–1700 / % | 1701–1709 / % | 1665–1709 / % |
|------|---------------|---------------|---------------|---------------|---------------|---------------|
| DJF  | -11.8 | -20.7 | -13.0 | -12.2 | -7.8 | -13.2 |
| MAM  | -23.1 | -22.5 | -11.3 | -4.9 | -6.5 | -12.6 |
| JJA  | -12.8 | -12.4 | 0.5 | 11.1 | 4.9 | -0.5 |
| SON  | -19.9 | -17.1 | -3.9 | -0.6 | -2.9 | -7.7 |

**Table 4.** Anomalies in Morin's *TCC* notes to the mean of present time series *TCC* (Meteoblue-Dataset; 1985–2014).





**Figure 8.** Total cloud cover (*TCC*) plotted in oktas of Louis Morin's readings from top to bottom: Yearly means of DJF, MAM, JJA, SON and the whole year. The box plots show the the seasonal and annual means of the time period of Morin's measurements and the mean of the data of Meteoblue from 1985–2014.





## 4.3 Direction of the movement of the clouds

**Figure 9.** Seasonal time series of the directional indices (DI: NI, EI, SI, WI in percentage of nonmissing days) from 1665 to 1709. The individual points indicate the mean per season of the direction of the movement of the clouds, the solid lines indicate the regression curves and the gray areas show the 95 % confidence interval.

To assess the atmospheric circulation during the LMM, we calculated the mean of Morin's sub daily notes, by vector addition. This was followed with our analysis of the data of the approach of Mellado-Cano et al. (2018). We then computed four





directional indices (DIs), one for each cardinal direction: northerly (NI) [315°, 45°], easterly (EI) [45°, 135°], southerly (SI) [135°, 225°], and westerly (WI) [225°, 315°]. They are defined as the percentage of nonmissing days per month with cloud motion from that direction (Wheeler et al., 2010; Mellado-Cano et al., 2018).

To illustrate the raw data, Fig. 9 shows the seasonal time series of the DIs from 1665 to 1709. The reduced number of just four indices allows us to easily interpret the main characteristics of the synoptic-scale circulation and the associated impacts in terms of temperature and precipitation. Thus, for example, an increased persistence of EI indicates dry advection from land, and the NI is related to cold advection from higher latitudes. An increased persistence of WI indicates wet advection from the ocean, and the SI is related to warm advection from lower latitudes. Furthermore, an increased persistence of WI in the

DJF season indicates warm advection from the ocean, whereas an increased persistence of WI in the JJA season indicates cold advection from the ocean. Apart from a high variability, a rise of the WI over the time period can be noted in each season. This conforms with earlier studies, which are based on ship logbooks (Wheeler and Suarez-Dominguez, 2006; Mellado-Cano et al., 2018). The DJF season indicates that the colder decade (1691–1700; cf. section 4.1) is marked by a strong EI and a weaker WI. In contrast, the decade from 1701–1709 is characterized by a dominating WI in the DJF season. This means that

the synoptic scale circulation could be an additional driving factor of the unusual cold conditions of the DJF season in 1691–1700 decade. The years from 1699 onwards are indicated by an unusual high percentage of WI, which seems to correlate to the warming. Lockwood et al. (2017) find that several factors are responsible for explaining the cold in the LMM. Thus, the cold could be explained by not only the reduced number of sunspots or a possible volcanic activity but also by possible other factors. Moreno-Chamarro et al. (2017) demonstrate that such exceptional wintertime conditions arose from sea ice expansion

and reduced ocean heat losses in the Nordic and Barents Seas, driven by a multicentennial reduction in the northward heat transport by the subpolar gyre. This would lead to blocking situations in winters, which would reduce the airflow from the western direction. This explanation is consistent with our results of the directions of the movement of the clouds.

Fig. 10 shows the seasonal DIs averaged for the LMM (1665–1709, color bars) and compares them with those of the reference period (1985–2014, gray bars). A comparison of different time periods can be seen, on the one hand, as a validation

of Morin's notes; while on the other hand it can show significant differences. Because Morin's observations are subjective, two issues arise by comparing the data with a contemporary time series: the different wind directions in the individual altitude layers and the unknown preference of which layer was chosen, if clouds appear at the same time in different layers. The second issue can be clarified by the assumption that the lowest appearing cloud layer was preferred, due to the fact, that from the point of view of an observer on the Earth's surface, nearer clouds seem to move faster and higher clouds can be hidden from

clouds in lower altitudes. Given that Hahn et al. (2001); Wu et al. (2011) investigated the cloud layer distribution and showed a higher percentage of low clouds in the JJA season, the first issue was handled by taking the mean DI of three pressure layers at 850 hPa, 700 hPa and 500 hPa for the DJF, MAM and SON seasons and by taking the mean DI of four pressure layers at 900 hPa, 850 hPa, 700 hPa and 500 hPa for the JJA season. We performed a two-tailed t-test and the result was that three indices show a significant difference between the two periods at the 90 % confidence level, namely SI and WI in the DJF

season and EI in the JJA season. So seasonally, winter is the season of the LMM that shows the largest (significant) difference with the reference period (Fig. 10). The winters were characterized by a lower frequency of WI and a higher frequency of EI.





The reduction of warm advection from the ocean leads to a cooling. In contrast, LMM summers exhibited a lower value of WI and EI, and an increase of SI and NI. Theoretically, these anomalies tend to favor warmer temperatures, but the the effects of the atmospheric circulation on European temperatures are weaker in summer than in winter because of the smaller pressure
gradients over the North Atlantic and the higher contribution of regional and thermodynamical processes (Vautard and Yiou, 2009). Nevertheless, in both cases, the reduced westerlies lead to more extreme seasons, with less influence of oceans. Hence, as Mellado-Cano et al. (2018) stated, on average, the role of the atmospheric circulation on European temperatures displayed a clear seasonal contrast: in winter, the dynamics favored cold conditions in Europe, while in summer they promoted warm conditions. In agreement, the temperature fingerprints of the DIs extend over larger European areas in winter than in summer
(Barriopedro et al., 2014). For both periods, the WI is the most recurrent DI, indicating the predominance in the region of westerly movement of the clouds. The frequency of days with meridional circulation (the sum of SI and NI) during the LMM was slightly higher than during the reference period, and the WI is characterized by lower frequencies in the LMM than in the reference period for all seasons. As shown by Barriopedro et al. (2014), a below-normal persistence of westerlies inhibits the warm oceanic advection over most of Europe during all seasons, except in summer, when it is rather associated with high-
pressure systems and radiative warming. In terms of precipitation, the WI is an optimal indicator for the transport of moisture fluxes to Europe, with decreased westerlies corresponding below average precipitation over large areas of central Europe all year round (Barriopedro et al., 2014). Accordingly, the reduced frequency of WI in Fig. 10 indicates a drier and colder DJF, a moderate MAM, JJA and SON in terms of precipitation and temperature compared to present (with the exception of summer). As seen earlier, this statement is true for DJF, MAM and JJA, but SON was cooler with respect to the reference period (Fig. 5).

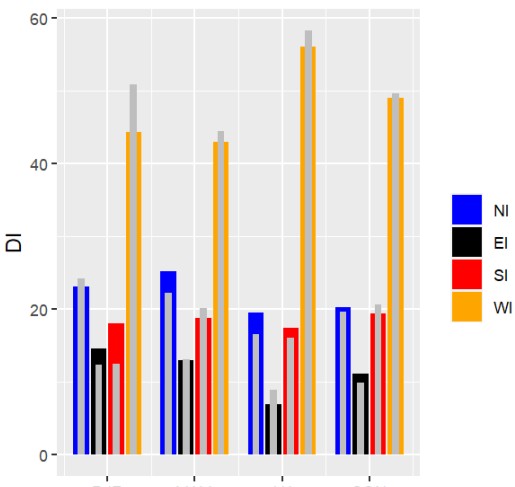

**Figure 10.** Seasonal frequencies of directional indices (DI: NI, EI, SI, WI in percentage of nonmissing days) averaged for the period of 1665–1709 (color bars). Gray thin bars indicate the corresponding values for the 1985–2014 reference period (Meteoblue Dataset).

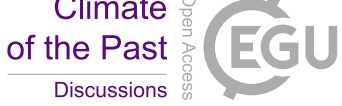

**Figure 11.** Visualization of DIs (in percentage of total days) as seasonal mean (DJF, MAM, JJA, SON), annual mean, outstanding DJF seasons and the January of 1709.



With the aim of analyzing the variability of the decades, we looked at the wind roses of the different seasons (Fig. 11). To ensure comparability, the mean values of Meteoblue (1986–2015) are also plotted. The strongest deviation from the current time shows the decade 1701–1709, which is characterized by high percentage values in W and WSW sectors. The decade 1691–1700 shows a different picture, where high percentage values in NNE, NE and ENE sectors are noted in the DJF, while

in the other seasons only small deviations from the current time are seen. It is also worth mentioning that the two decades 1671–1690 show a strong decrease in air flows from the westerly direction. In addition, a strong interdecadal variability can be seen. An interesting picture is shown by the wind roses of the exceptional winters in the LMM. The winter of 1679 is characterized by low W values and high values in N and NE direction. The winter of 1694/95 shows a similar picture, whereas the winter of 1683/84 shows a moderate distribution but with a pronounced E direction. The January of 1709 is characterized

by a strong NNE direction and weaker expressions of ENE, E, SSE and WNW. So, the January shows a circulation structure, which leads to cold temperatures. Whereas, looking on the whole winter of 1708/09, the dominant direction of movement of the clouds is the WSW direction with a percentage of 36 %. This illustrates the contrast of the individual months in the winter of 1708/09 with respect to the circulation situation and, subsequently, the temperatures.

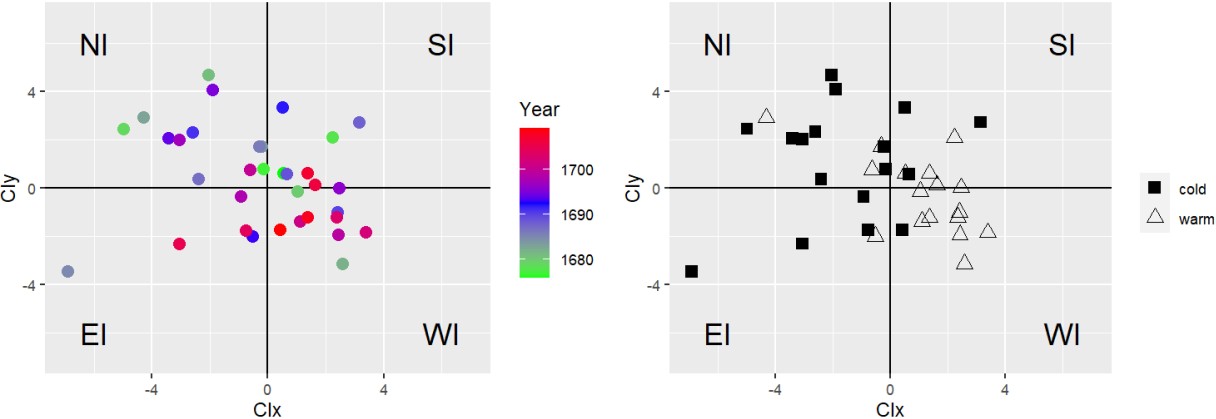

**Figure 12.** (a) Scatterplot of the CI for the LMM winters, with colors indicating the year within the LMM. The x axis represents the *CIx* coordinate of the CI and the y axis the *CIy* coordinate of the CI. (b) As in (a) but showing two different clusters, where squares are below the Median of $T_{mean}$ and triangles are above the Median of $T_{mean}$.

To characterize the interannual circulation variability within the LMM and synthesize the four-dimensional information of the DIs, we computed a cumulative circulation index (CI, Mellado-Cano et al. (2018)) that aggregates the standardized values $(DI_{st} = (DI_{Morin} - DI_{Ref})/std(DI_{Ref}))$ of the four DIs in two components (*CIx,CIy*):

$$CIx = WI_{st} + SI_{st} - (NI_{st} + EI_{st})$$

$$CIy = NI_{st} + SI_{st} - (EI_{st} + WI_{st}).$$

The component *CIx* is purely based on atmospheric circulation but it can also be used as an indicator of the European tem-

perature conditions that could be expected from the dynamics, with positive values of *CIx* indicating an overall warming and





negative values of *CIx* indicating an overall cooling. *CIy* measures the degree of meridional (NI, SI) versus zonal (WI, EI) circulation, with positive values indicating a dominance of the former and with negative values indicating a dominance of the latter. The results are shown in Fig. 12 and the standardized time series of DI separated to the seasons can be seen in the additional material (Fig. A3–A6). The left-hand plot shows a scatterplot with a color marking of the different years. Again, the

decade 1701–1709 (red dots) shows a shift towards WI, while the decade 1691–1700 is relatively scattered, indicating high variability but with a tendency towards NI. In the right-hand panel, squares represent cold winters and triangles represent warm winters. The separator of the warm and cold winter seasons is the Median of the Morin time series. This plot can be seen as an internal validation of the Morin data. Warm winters imply a tendency towards WI, while cold winters show a tendency towards NI. The relationship of WI and mean temperature can be read in Fig. A7, which is approximately 1° per 10 %. This shows that

large-scale weather conditions can explain at least part of the cooling in the winter months.

## 5   Conclusions

We have digitized three meteorological variables (i.e., temperature, direction of the movement of the clouds and cloud cover) from copies of Louis Morin's original measurements (source: Institute of History / Oeschger Centre for Climate Change Research, University of Bern) and subjected them to quality analysis to make these data available to the scientific community.

Our available data cover the period 1665–1709 (temperature beginning in 1676). Thus, during this time period, three values are available on a daily basis for each variable: morning, midday/afternoon, and evening. Hypotheses about largely unavailable metadata were supported with statistical tests, as well as by drawing on proxy data. Only small irregularities were uncovered in the data and they are therefore considered trustworthy, with respect to the possibilities at this time. In addition, Morin's measurements have a high correlation with grape harvest dates (*GHD*) and show a snowfall frequency threshold of 1.5 °C,

which is an indicator of the lower calibration point of the thermometer. An analysis of the variables examined here shows internal consistency and thus may be useful in further describing the Late Maunder Minimum (LMM).

The LMM is known as an exceptionally cold period of the Little Ice Age (LIA). However, seasonal temperatures reveal a more differentiated picture of a high frequency of cold winters and falls but is close to modern values in springs and summers (see Fig. 5). The highest numbers of cold winters occur in the decade 1691–1700. Previous studies show that the main causes

include volcanic activities, the reduction of the total solar irradiance (decrease of sunspots), or a a multicentennial reduction in the northward heat transport by the subpolar gyre. However, at least additional attribution (or possibly an implication of the latter) can be seen in the following: in this period, winter months show a significant lower frequency of westerly direction of movement of the clouds (see Fig. 10). This reduction of advection from the ocean leads to cooling in Paris in winter. This can be seen very strongly when comparing the last decade of the 17[th] century (cold) and the first decade of the 18[th] century (warm)

(See Fig. 9). A drop of the frequency of the westerly index (WI) between 1691–1700 is followed by a rise of WI in the first decade of the 18[th] century. A lower frequency of westerly direction of movement of the clouds can also be seen in summer and this reduction probably leads to moderate/warm temperatures in summer. Thus, unusually cold winters in the LMM can be partly attributed to a lower frequency of westerly direction of movement of the clouds, and a rise of either the northerly index




(NI) or easterly index (EI). This is shown in Fig. 12, where outstanding cold winters coincide with a low WI. Interestingly, the

notorious winter of 1708/09 is not particularly pronounced in terms of mean temperature because the December of 1708 was moderate and devastating cold temperatures (-18 °C) were restricted to January. This contrast is also reflected in the analysis of the movement of the air, where the WSW direction is dominant for DJF 1708/09 and NNE and E for January 1709 and so, in terms of temperature, a moderate December and February weakens indices concerning the whole winter of 1708/09.

An impact analysis reveals that the winter of 1708/09 was indeed a devastating winter with respect to consecutive ice days.

Although in terms of consecutive cold, days the winters of 1676/77, 1678/79, 1683/84, 1692/93, 1694/95 and 1696/97 are more pronounced. The absolute number of cold days and ice days is highest in the last decade of the 17th century (exceptional winters: 1690/91, 1691/92, 1694/95 and 1696/97).

The analysis of the cloud cover reveals the highest total cloud cover (*TCC*) in the first decade of the 18th century, whereas outstanding high values appear also in 1693 and 1694. Interestingly, all seasons show an increasing linear trend over the whole

period. However, further studies are required to validate and to further explain this trend. Nevertheless, the investigation of the cloud cover data revealed a high discrepancy of the seasons, where DJF (-13.2 %) and MAM (-12.6 %) show a negative anomaly of *TCC*, whereas JJA (-0.5 %) show a moderate anomaly of *TCC* with respect to the 30 year mean of the Meteobluedata (1985–2014). All in all, Morin's measurements are characterized by an extraordinary durability and are therefore a good source to get a comprehensive picture of the regional climate in the LMM. Therefore, we will investigate some further variables (precipitation

and humidity), which will provide further insights on the climate in Paris during the LMM.

*Data availability.* Yes





**Appendix A**

**A1**

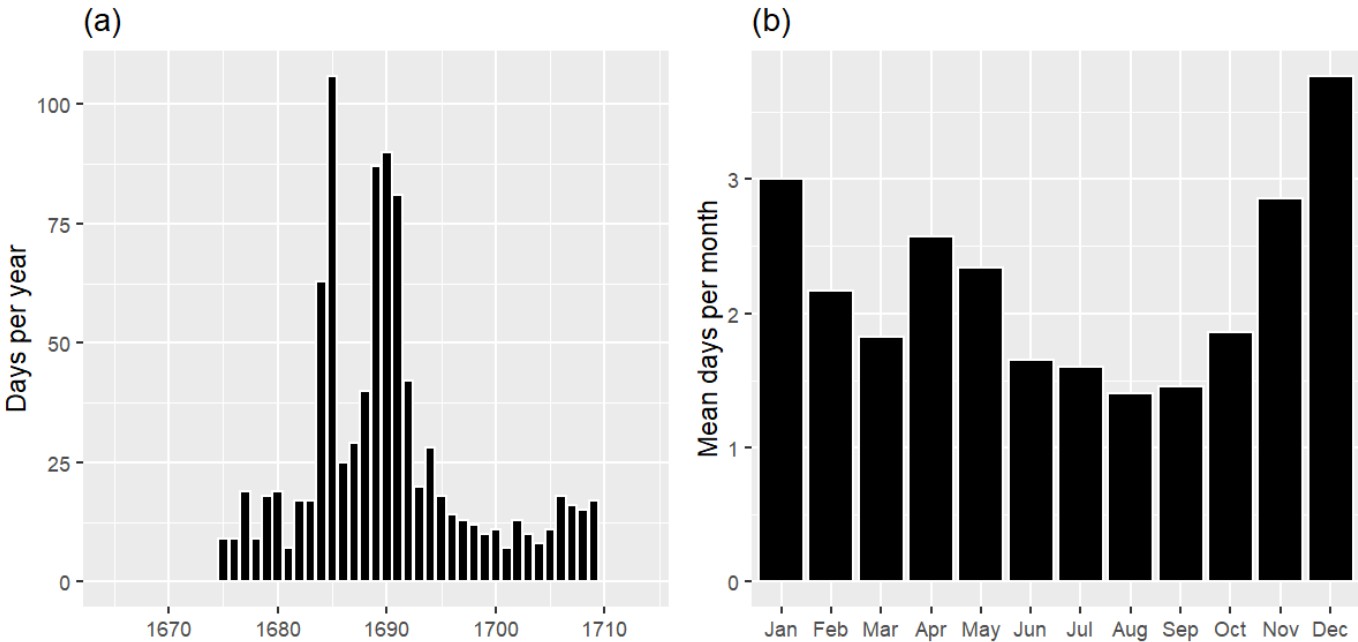

**Figure A1.** In (a) the total amount of days, when the evening temperature exceeds the midday/afternoon temperature is shown. In (b) the frequency of days, when the evening temperature exceeds the midday/afternoon temperature, per month is shown.



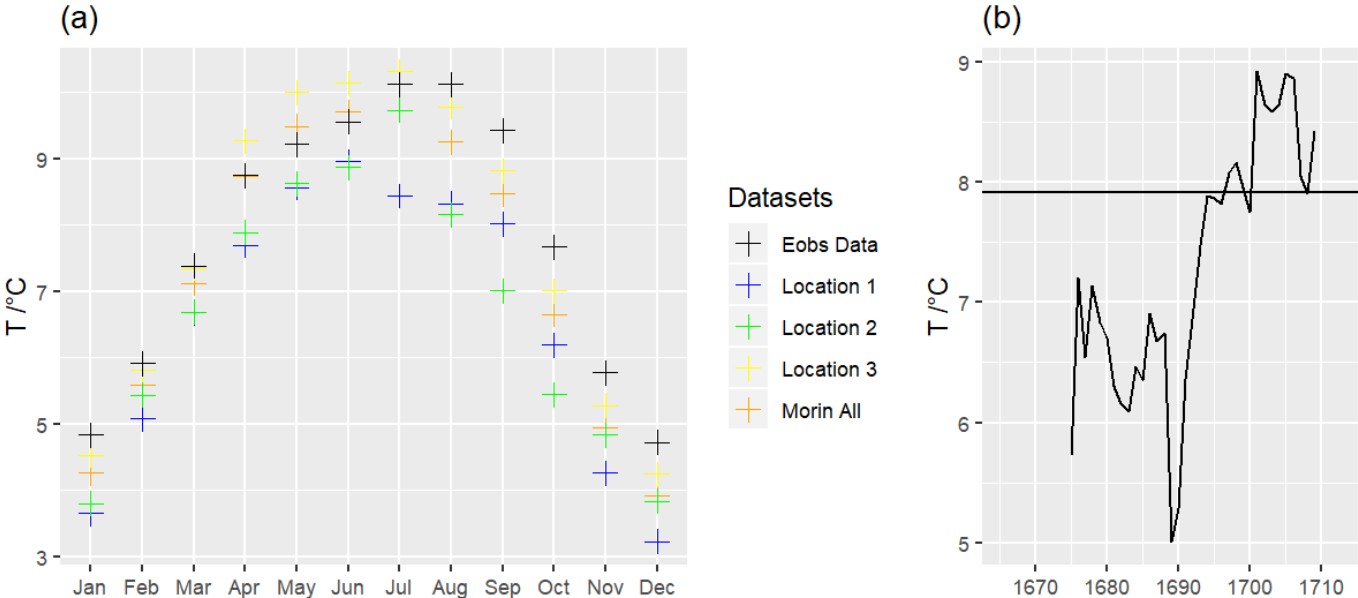

**Figure A2.** In (a) the mean of *DTR* per month of the E-Obs data, of the different locations and of the whole time period can be seen. In (b) the yearly mean of *DTR* is plotted and the horizontal line represents the mean of the reference period from 1961–1990 (E-OBS).





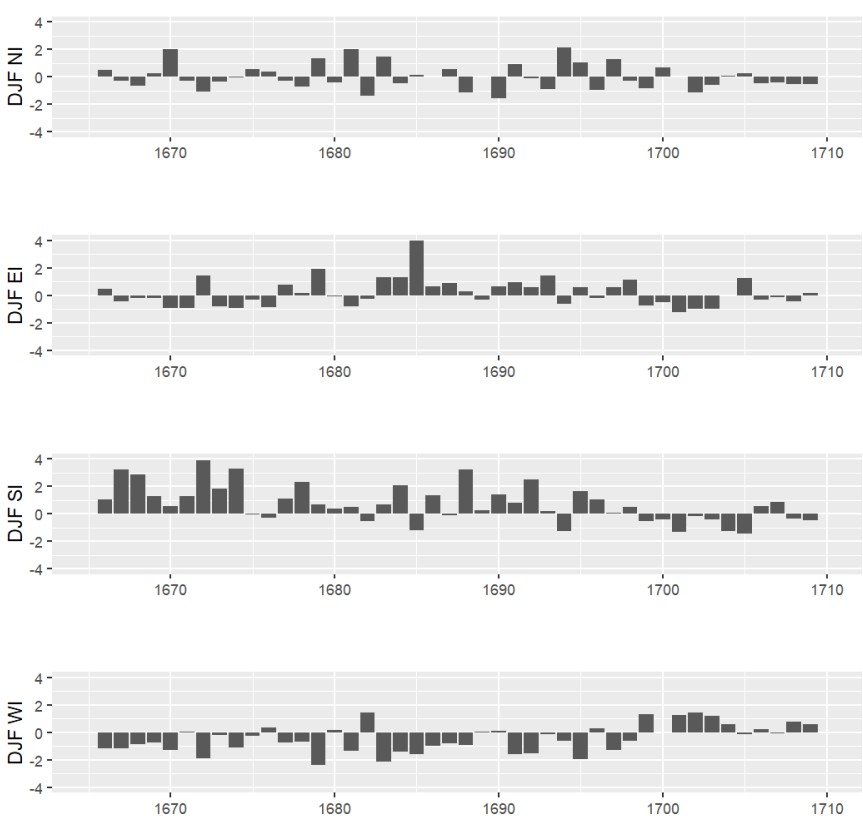

**Figure A3.** Standardized DJF DIs time series for the LMM (1665–1709) with respect to the mean of the reference period of 1986–2015.

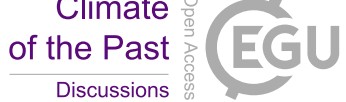

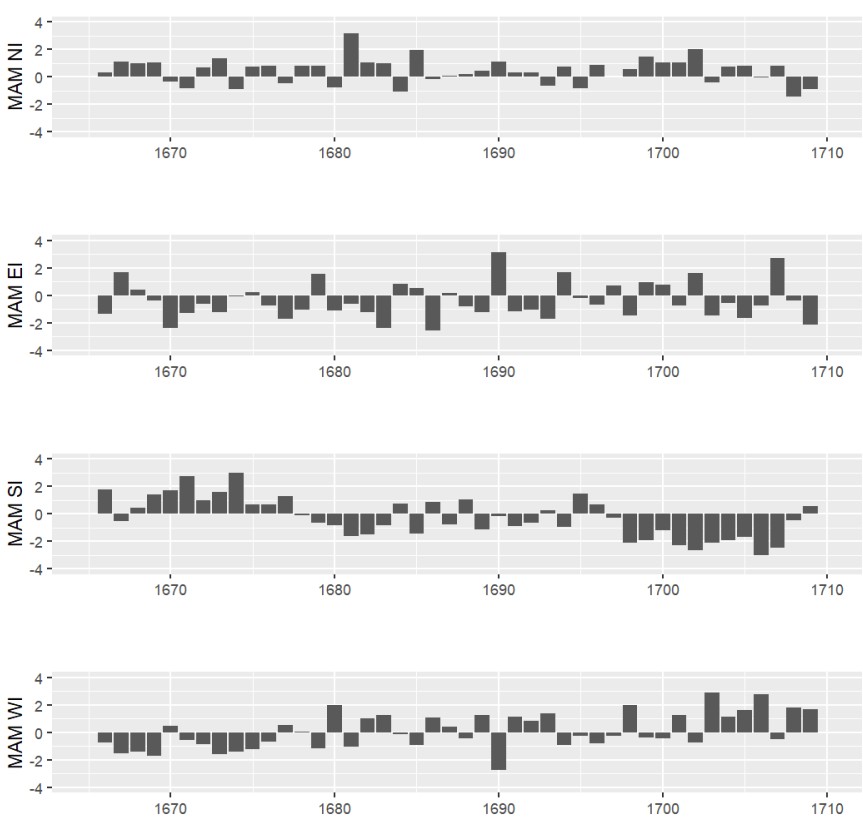

**Figure A4.** Standardized MAM DIs time series for the LMM (1665–1709) with respect to the mean of the reference period of 1986–2015.



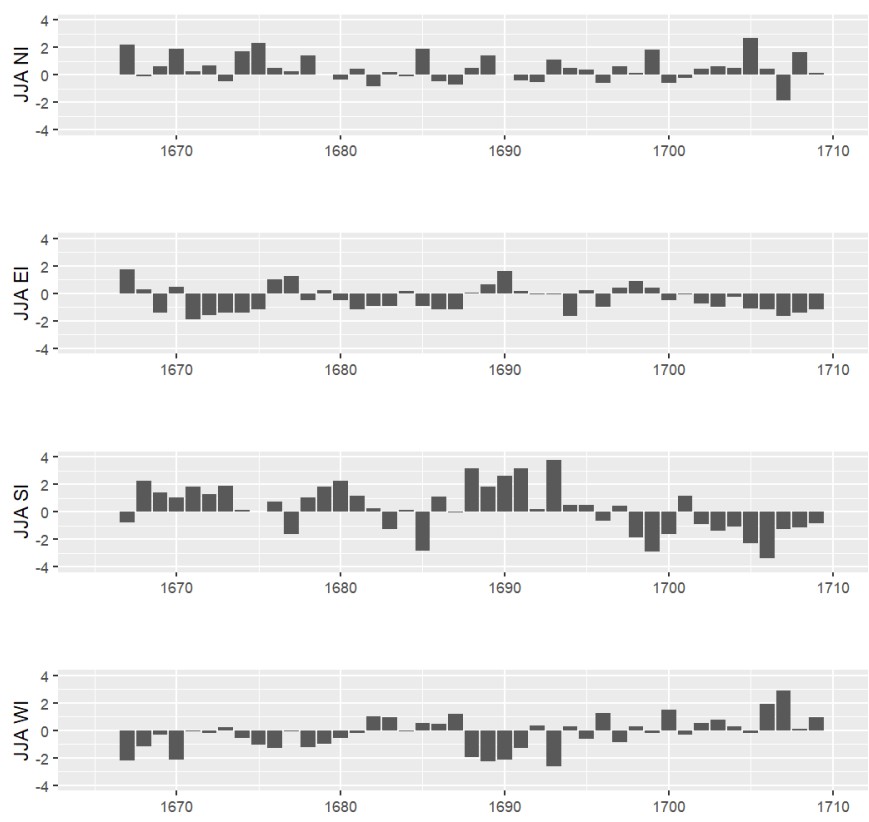

**Figure A5.** Standardized JJA DIs time series for the LMM (1665–1709) with respect to the mean of the reference period of 1986–2015.

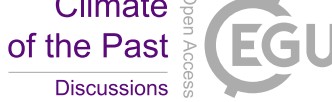

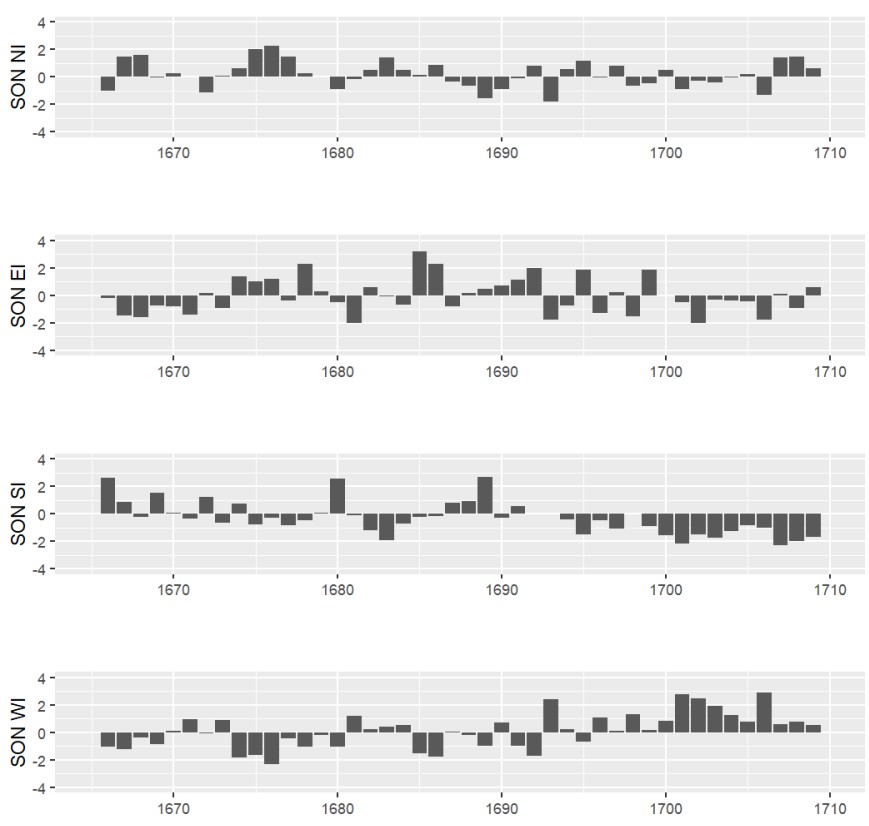

**Figure A6.** Standardized SON DIs time series for the LMM (1665–1709) with respect to the mean of the reference period of 1986–2015.





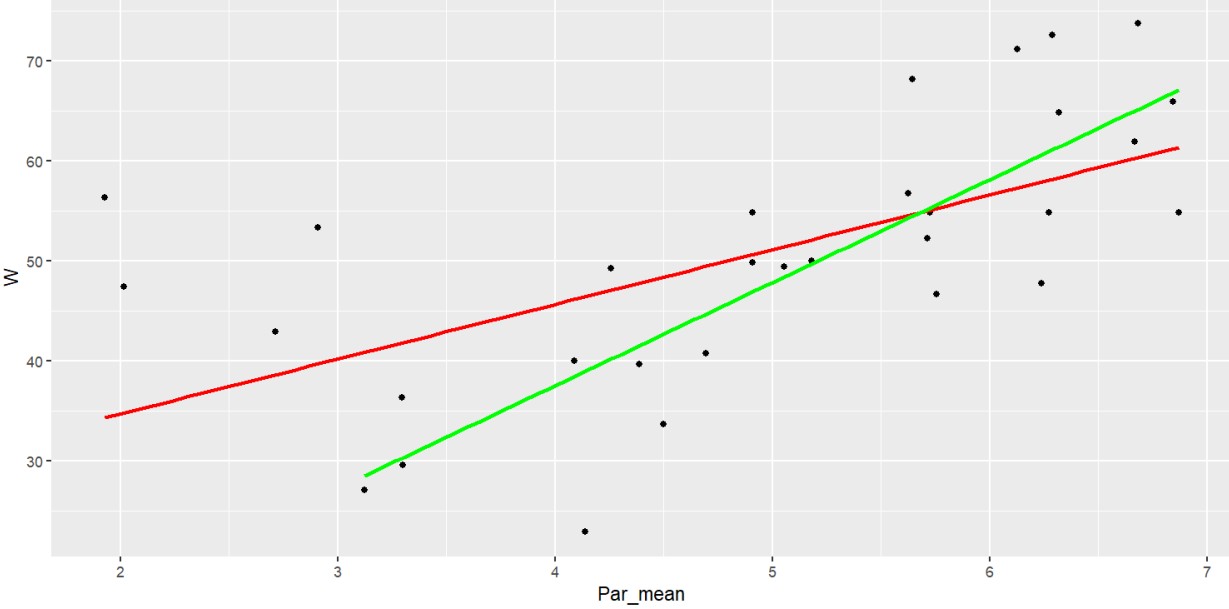

**Figure A7.** The correlation of the WI to the mean temperature in Paris (Meteobluedata; 1985–2014), a linear regression (red line) over all data points and a linear regression (green line) excluding outstanding/extraordinary years (1985, 1986, 1987 and 2010).

*Author contributions.* Christian Rohr and Christian Pfister provided both expertise and access to copies of Louis Morin's measurements.

Furthermore, during my scientific stay in Bern, I was able to share their knowledge of climate history. Ulrich Foelsche, as my supervisor, was permanently available for questions during the development process and contributed significantly to the content of this paper, as well as to its scientific correctness.

*Competing interests.* The authors declare no competing interests.

*Acknowledgements.* This paper has been supported by the Austrian Science Fund Project (Climate History of Central Europe During the

Little Ice Age; Clim_Hist_LIA P31088-N29).





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
