# Peer review of "Subdaily meteorological measurements of temperature, direction of the movement of the clouds, and cloud cover in the Late Maunder Minimum by Louis Morin in Paris"

_Climate of the Past, 2021_

## Author Response (AR2)

*The 'late Maunder minimum' is not a period of climate*

*I strongly suggest that the authors reconsider their use of the term "late Maunder Minimum" as a name describing a period of climate.*

*The 'Maunder minimum' is a term first coined by J.A. Eddy, [1] to describe the period in which the Sun had a prolonged sunspot minimum, first noticed by E.W. Maunder in the 1890's. The Maunder Minimum is now generally used to describe the period of very low observed sunspot numbers between 1640 and 1720.*

*The 'Maunder minimum' it is not a climatic term, and it is extremely misleading to use it as such. To do so suggests an a-prior belief that any climate changes observed are due to lower solar activity.*

*The authors say that the "late Maunder Minimum" could be due to a number of different factors, (lines 352-357 and 429-432), including sunspots, volcanic activity and ocean heat transport. I am sure the authors did not mean to imply that volcanoes and the ocean influence the Sun, but this demonstrates the confusion that can be caused by using an astronomical term for solar activity as a description for a period of climate.*

*The authors state that "Lockwood et al. (2017) find that several factors are responsible for explaining the cold in the LMM." (line 352). This is not correct. Lockwood (2017) [2] does not use the term 'LMM'/'Late Maunder minimum'. Neither does the study use 'Maunder minimum' as a term to describe a period of cool climate.*

*The term 'Late Maunder minimum' has been used incorrectly in a few published climate studies (referenced by the authors), but that should not be used as a precedent for its continued incorrect use.*

*If the authors need a name for the climate period 1675 to 1715, they should consider creating a neutral name that does not imply a cause or exaggerate the climatic conditions at the time.*

*[1] Eddy, J.A.,  The Maunder Minimum, Science, 1976*
*[2] Lockwood, Owens, Hawkins, Jones and Usoskin, Frost fairs, sunspots and the Little Ice Age, Astronomy and geophysics, 2017*

Thanks for the comment, which raises an important point. We think that it is difficult to use terms that are defined by climatology. The period we are studying lies in the Little Ice Age (1300–1850), which, however, stretches over centuries and is also subject to some degree of criticism. Subperiods in the Little Ice Age, which are defined by climatic parameters, do not exist. The reason is obvious because a definition with climate-relevant parameters is difficult. Cold periods and warm periods are heterogeneous in terms of locality and temporal occurrence as well as influenced by many factors. I.e. a temporal limitation is difficult. Thus, there is another possibility to borrow or include a term

from another discipline. Strictly speaking, even using the term Middle Ages would be a borrowed term, since it is not defined by climatic parameters in beginning and end. What we want is simply to use a term that is known in our discipline and allows a quick attribution for the reader. We want to use this term neutrally with respect to climate-relevant parameters and will also correct the paper where it could lead to misunderstandings, as well as state this explicitly. So, we want to keep the term and show by the following list of publications that this term is already established.

Alcoforado, M.-J., M. de Fátima Nunes, J. C. Garcia, and J. P. Taborda, 2000: Temperature and precipitation reconstruction in southern Portugal during the Late Maunder Minimum (AD 1675–1715). Holocene, 10, 333–340, https://doi.org/10.1191/095968300674442959.

Barriendos M. Climatic variations in the Iberian Peninsula during the late Maunder Minimum (AD 1675-1715): an analysis of data from rogation ceremonies. *The Holocene*. 1997;7(1):105-111. doi:10.1177/095968369700700110

Barriopedro, D., Gallego, D., Alvarez-Castro, M.C. et al. Witnessing North Atlantic westerlies variability from ships' logbooks (1685–2008). Clim Dyn 43, 939–955 (2014). https://doi.org/10.1007/s00382-013-1957-8

Luterbacher, J., Rickli, R., Tinguely, C., Xoplaki, E., Schüpbach, E., Dietrich, D., Hüsler, J., Ambühl, M., Pfister, C., Beeli, P., Dietrich, U., Dannecker, A., Davies, T., Jones, P., Slonosky, V., Ogilvie, A., Maheras, P., Kolyva-Machera, F., Martin-Vide, J., Barriendos, M., Alcoforado, M., Nunes, M., Jónsson, T., Glaser, R., Jacobeit, J., Beck, C., Philipp, A., Beyer, U., Kaas, E., Schmith, T., Bärring, L., Jönsson, P., Rácz, L. and Wanner, H. (2000), Monthly mean pressure reconstruction for the Late Maunder Minimum Period (AD 1675–1715). Int. J. Climatol., 20: 1049-1066. https://doi.org/10.1002/1097-0088(200008)20:10<1049::AID-JOC521>3.0.CO;2-6

Luterbacher, J., Rickli, R., Xoplaki, E. et al. The Late Maunder Minimum (1675–1715) – A Key Period forStudying Decadal Scale Climatic Change in Europe. Climatic Change 49, 441–462 (2001). https://doi.org/10.1023/A:1010667524422

Mellado-Cano, J., Barriopedro, D., García-Herrera, R., Trigo, R. M., & Álvarez-Castro, M. C. (2018). Euro-Atlantic Atmospheric Circulation during the Late Maunder Minimum, *Journal of Climate*, *31*(10), 3849-3863. Retrieved Mar 16, 2022, from https://journals.ametsoc.org/view/journals/clim/31/10/jcli-d-17-0261.1.xml

Niedzwiedz, T., 2010: Summer temperatures in the Tatra Mountains during the Maunder Minimum (1645–1715). The Polish Climate in the European Context: An Historical Overview, R. Przybylak et al., Eds., Springer, 397–406, https://doi.org/10.1007/978-90-481-3167-9.

Rácz, L., 1994: The climate of Hungary during the Late Maunder Minimum (1675–1715). Climatic Trends and Anomalies in Europe 1675–1715, B. Frenzel, C. Pfister, and B. Gläser, Eds., G. Fischer, 43–50.

Wanner H, Pfister C, Bra´zdil R, Frich P, Frydendahl K, Jo´nsson T, Kington J, Lamb HH, Rosenørn S, Wishman E (1995) Wintertime European circulation patterns during the Late Maunder Minimum cooling period (1675–1704). Theoret Appl Climatol 51:167–175. doi:10.1007/BF00867443

Xoplaki, E., P. Maheras, and J. Luterbacher, 2001: Variability of climate in meridional Balkans during the periods 1675–1715 and 1780–1830 and its impact on human life. Climatic Change, 48, 581–615, https://doi.org/10.1023/A:1005616424463.

Zinke, J. , Dullo, C. , von Storch, H. , Müller, B. , Zorita, E. , Rein, B. , Mieding, B. , Miller, H. , Lücke, A. , Schleser, G. , Schwab, M. , Negendank, J. , Kienel, U. , Ruoco, G. and Eisenhauer, A. (2004): Evidence

for the climate during the Late Maunder Minimum from proxy data available within KIHZ , The climate in historical times : towards a synthesis of holocene proxy data and climate models / [GKSS-Forschungszentrum]. Hubertus Fischer ... (eds.) Berlin [u.a.] : Springer, S. 397-414 (GKSS School of Environmental Research), ISBN: 3-540-20601-9 .

Zorita, E., Von Storch, H., Gonzalez-Rouco, F. J., Cubasch, U., Luterbacher, J., Legutke, S., et al. (2004). Climate evolution in the last five centuries simulated by an atmosphere-ocean model: global temperatures, the North Atlantic Oscillation and the Late Maunder Minimum. *Meteorologische Zeitschrift, 13*(4), 271-289.

Furthermore, we will change the above-mentioned citation, which was actually not quoted correctly by us.

Dear reviewer,

we thank you for the careful read and helpful comments to improve the paper. The comment aims to identify some points for improvement as well as more fundamental methodological errors. The latter point in particular seems to us to require explanation, justification or revision by us. For ease of reading, we will list each comment in italics as well as attach a response directly below each comment. Citations that are not listed in the paper are listed at the end.

*The paper describes a meteorological record covering the late 17th and early 18th century for Paris. Given the length of the record for a period in which instrumental records are very rare, the data and results presented are relevant to the scientific community. However, in my view the manuscript presents major flaws in terms of how the data are analysed and the results interpreted.*

*General comments:*

*C1.1- The comparison with modern data is misleading, as the historical data are affected by unknown biases related to the measurement location, thermometer scale, and other factors. These biases can easily surpass climate variability in magnitude. I understand the appeal of comparing historical extreme climate events with modern climate, but this is not possible in a scientifically sound way without applying the necessary bias adjustments - which is very hard if not impossible in this case given the lack of metadata. This is particularly true for temperature and cloud cover.*

In the case of temperature, we have made an effort to validate both internally and externally with proxy data such as wine harvest dates. The reason for this was that the calibration of the thermometer by Legrand & Le Goff (1987) was based on proxy data indicating a similar climate of a time period in the 19th century and thus the temperature measurements of this time period served as calibration. Thus, we tried (p.10-13) to show possible inhomogeneities. The results show satisfactory results considering that we deal with measurements in the 17th and early 18th centuries. Nevertheless, we will point out explicitly that biases will remain to a certain extent. Without metadata it is difficult to quantify those biases, but with our analysis it can be seen that the temperature measurements are consistent with the hypothesis of earlier studies. In terms of quantity the time series is mostly inconspicuous. We will display inhomogeneities and provide a time series of the homogenized data in a data file (See C2.9 and C2.13 of the second reviewer).

The graphical illustration of the mean value of the present time in Fig. 8 (Cloud cover) is only intended as a comparative value. A comparative value, which does not establish a statement of the climate variability of the different time periods, but shall only show, whether the

subjective "measurement" of the cloud cover results in plausible values. This is true for temperature as well. It is for instance easier to identify cold winters showing anomalies (Fig. 5) as when showing absolute values. But again, we do not claim that for instance a difference of 0.5 °C shows a different in climate between those periods. But we want to show that cold winters, with anomalies up to -4 °C, appeared frequently in the analyzed time period.

Therefore, we feel justified in maintaining these comparisons, with clearer explanation.
* * *
*C1.2- In addition to the previous point, the Meteoblue dataset used by the authors is a commercial product that, as far as I know, has never been evaluated in peer-review literature. If that is true, it should not be used for a scientific article. My suggestion is to drop the use of a modern reference period for temperature and cloud cover - except perhaps to assess data quality such as for the NDR calculation - and concentrate on the decadal variability of the studied period.*

We adopted the ERA5 reanalysis instead of the Meteoblue dataset, because it is obvious that the former dataset is far more often used in the scientific literature. Nevertheless, as stated in C1.1 we want to keep our "comparison" with the reference period.
* * *
*C1.3- The temperature record is clearly affected by inhomogeneities. The authors actually do a very good job in pointing them out, by mentioning relocations, changes in the temperature scale, and changes in the ventilation of the instrument (Tab. 1; also Fig. 1a and 1b point to at least two important inhomogeneities). However, this fact is completely ignored when analysing the data. There are some confusing sentences about this at the end of Sect. 3.3 that actually raise even more doubts about the quality of the data. I believe that some kind of statistical homogenization is necessary, even though reference series for this period are scarce. Beside the Central England Temperature series, there exist many temperature reconstruction that could be used.*

We will be content to only show anomalies and provide both time series of the raw data and a homogenized time series in one data file. However, since the homogenization does not affect the main statements in the paper, we want to keep the graphs of temperature based on the raw data after calibration.

Regarding a statistical homogenization: Proxy data do not seem to us to be useful since they do not have the required temporal resolution. However, a comparison has already been discussed in the paper anyway with grape harvest dates. A homogenization with the CET is conceivable, but also connected with problems. First, the CET has only a monthly resolution and is based on mean temperatures, whereas our data have a daily resolution, and we are interested in the maximum and minimum temperatures. A homogenization with only one time series can further lead to spurious correlations and climate signals of the own measurements can be lost.

Furthermore, comparisons with other measurements show quite consistent results (See below for instance the winter temperature anomalies of De Bilt, CET and Paris).

[Figure]

*Figure 1: DJF Anomalies of Paris, De Bilt and CET (Manley, 1974)*

*C1.4- How does this record relate to the widely available long monthly temperature series for Paris? Is that series also based on Morin's observations? Are there any differences from your data?*

If this point refers to the temperature series of Rousseau (2009, 2013) then yes. The difference is that we wanted to provide data on a daily basis.

*C1.5- Many equations and definitions appear in the results. They should be moved to the methods section.*

We will make slight modifications but would like to keep most of the layout due to readability.

*C1.6- I am not a native English speaker but the quality of the language seems rather poor to me, to the point that I had difficulties understanding some sentences.*

The article has been proofread by PRS. (https://www.proof-reading-services.org/en/)

Specific comments:

*C1.7- The procedure to convert the temperature readings to Celsius need to be explained more in details, since the given references are in French. Besides, the conversion formulas (Eqs. 1-3) are not completely clear to me: I would expect that the TM in the three equations refer to different observation times, but this is not indicated. Moreover, it is often mentioned in the manuscript that the thermometer was filled with spirit: is this an assumption or a known fact? How do you explain that a linear conversion does not introduce a bias at high temperatures?*

A more detailed explanation of the conversion to °C will be adopted. Also, the terminologies (TM) will be pointed out more clearly. TM basically means measured temperature and refers to the highest measured value per day when calculating Tmax and to the lowest measured value per day when calculating Tmin.

We do not know for sure if the thermometer was filled with mercury or spirit. Earlier literature assumes the latter. Nevertheless, we could not identify biases which can appear in temperature measurements using spirit (See Camuffo, 2016). We will make this point clear. However, the term "liquid" was used just once in the context of the calibration procedure, which is indeed an assumption. We will modify the text to avoid misunderstandings.

An unreasonably high bias at high temperatures due to radiation, liquid thermometers, etc. would be visible when comparing the GST (growing season temperature) with grape harvest dates and when, what we did but don't show it in the preprint, looking at qq plots with EOB-S data (Original units of Morin and temperature series of EOB-S). The comparison is not scientifically valid, but unreasonably high biases could have been seen there, if existing.

*C1.8- Equation on page 10 (number missing): I believe the indices i,j,k here are in the wrong positions.*

Thank you for making the effort to understand the equation and to point out the misleading indices. We changed that and added a number to the equation.

*C1.9- P11, L207: Dai (2006) shows that the effect of pressure on snowmelt is negligible in the lower troposphere. Besides, increasing humidity cause the melting point temperature to decrease (i.e. a lower temperature is required for snow), not increase. More importantly, precipitation phase at the surface depends on the temperature profile above the station, of which surface temperature is merely a proxy (e.g. it can be significantly warmer 1 km above the surface than at the surface, hence it can rain with negative temperature). Another important factor is precipitation intensity (higher intensity implies higher melting point temperature).*

We are aware that there are different factors which lead to a different melting point. Therefore, we see the calculation in Fig. 3 as a statistical approach to validate for plausibility. Or in other words: To check for inhomogeneities of the temperature measurement near the snow/rain threshold.
* * *
*C1.10- P11, L215: "So, if..." - something wrong with this sentence, snowfall frequency is not measured in °C.*

Thank you. We correct the sentence.
* * *
*C1.11- Equation on page 12 (number missing): What is the factor 2 for? The notation for the sums is confusing.*

The factor 2 will be deleted, because it appears in the denominator and in the numerator. But we will keep the notations of the sum (See also Camuffo, 2017)
* * *
*C1.12- P12, L231: What is a typical value for NDR for data measured indoor? How relevant is the change from 0.8 to 0.95 in 1688?*

For indoor measurements the NDR value is below 0.3 (See Electronic supplementary material Camuffo, 2017). Therefore, we can state that the measurements were performed outside. A change of 0.8 to 0.95 is of relevance, because it points out that the exposure or maybe the thermometer itself changed. (See also C2.13 for more detail)
* * *
*C1.13- P13, L263: The Maunder and Spörer Minima are defined by solar activity, not by climate, and the influence of solar activity on climate is still uncertain. This is mentioned briefly later in the manuscript, but I believe it should be clarified already in the introduction. The choice of LMM to describe the period covered by the data is perhaps not the best as it gives the impression that the climate anomalies were mainly driven by solar activity.*

We adopt a clearer explanation of the terminologies but point out once again that the terminology "Late Maunder Minimum" is well accepted in historical climatology.
* * *
*C1.14- P22, L349: "This means that..." - Circulation is an essential requirement for cold winters rather than an additional driving factor. Even a possible solar influence would mainly act through changes in circulation (e.g. Barriopedro et al., 2008).*

We totally agree to this comment and will modify statements, which are misleading or stating too strong hypotheses. What we have found in our study is just a lower frequency of WI (basically frequency of clouds moving from west to east) and that there is a correlation of a

low WI with lower temperatures in winter. This can also be seen as a further validation of the temperature measurements.
* * *
*- P22, L367: How is exactly the DI calculated from modern data? Are clear days excluded? If not, there would be an obvious bias with respect to Morin's observations. How dependent are the results from the choice of the levels? This comparison should be done using an open, peer-reviewed dataset (e.g. ECMWF ERA5 reanalysis), or dropped.*

Thank you for coming up with this point. We did not exclude clear days in the first plot, but changed that and the results are still the same. A different weighting of the levels would result mainly in slightly different numbers of EI and SI. WI has in all levels of our interest higher values than our calculated mean for WI of Morin's measurements. We excluded the clear days (TCC<10%) and the result for ERA5 can be seen in the following below.

However, we will reconsider slightly different levels for comparison. In the plot below we took the mean of the 900 hP, 850 hP, 700 hP and 500 hP level for summer and spring and the same for winter and autumn with a weight on the 900 hP level of the factor of 2.

We would like to point out in particular the agreement of Morin's observations with the reanalyses, which is outstanding considering that the measurement was made by eye.

[Figure]

*Figure 2: Seasonal mean of the DI of Morin's measurements (colored) and ERA5 reanalysis (1980-2020)*

Brandsma, T. and van der Meulen, J.P. (2008), Thermometer screen intercomparison in De Bilt (the Netherlands)—Part II: description and modeling of mean temperature differences and extremes. Int. J. Climatol., 28: 389-400. https://doi.org/10.1002/joc.1524

Manley, G. (1974), Central England temperatures: Monthly means 1659 to 1973. Q.J.R. Meteorol. Soc., 100: 389-405. https://doi.org/10.1002/qj.49710042511

van der Meulen, J.P. and Brandsma, T. (2008), Thermometer screen intercomparison in De Bilt (The Netherlands), Part I: Understanding the weather-dependent temperature differences). Int. J. Climatol., 28: 371-387. https://doi.org/10.1002/joc.1531

Dear reviewer,

we thank you for the careful read and helpful comments to improve the paper. We feel we can address the main points of the comment and think that the paper has gained in quality because of the modifications. For readability, we have numbered and italicized the individual comments. Our response immediately follows the individual comments.

*The authors have carefully examined a part of the measurements contained in Louis Morin's observation notes kept from 1665 to 1713. These notes are of great importance, given the scarcity of measurements at that time, for understanding the climate during "The Little Ice Age". Temperature and pressure measurements have been the subject of previous studies, well mentioned by the authors. We believe that the interest of the article relates to:*

*- the confirmation of the high quality and reliability of Morin's temperature measurements.*

*- the confirmation of the characterization, using these, of the climate of this period*

*- the study of observations of the direction of the wind deduced from the movement of clouds, which had not yet been the subject of previous studies.*

*This last point allows the authors to establish a link between the temperature differences (in particular the remarkably very cold winters-springs, but also the summers-autumns close to modern values) with the characteristics of the atmospheric circulations.*

*The abstract gives a precise idea of the content and of the conclusions of this study.*

*C2.1-Morin's Manuscript, which can be consulted at the Institut de France and has already been used by several of the authors cited in reference, includes observations up to 1713. We are surprised that the authors of the article could not use a complete copy of Morin's manuscript or even failing that, did not use in their analysis the data from 1710 to 1713 published in volume 2 of the 1992 note by Legrand and Legoff, cited in the article.*

We had access to the Legrand and Le Goff book, of course, but we wanted to base the study on the original data. I.e. in this book, only processed ones are published. In the case of cloud cover and direction of the movements of the clouds they published data only up to 31 December 1709 in tabular form.

However, after your comment I consulted the Institut de France and added the data from 1710 to 1713. Thank you for the hint. After consultation with the director of the Institut de France, we are allowed to use the data for the analyses, but in order to publish them day per day in tabular form, we have to wait for a meeting on May 17, 2022. Until then, we can only make the daily data public in tabular form up to and including 1709.

*C2.2-To be more in conformity with the title of the article, it would be desirable to include these data de 1710 à 1713 in the study or otherwise to modify the title of the article in : "Subdaily meteorological measurements of temperature, direction of the movement of the clouds, and cloud by Louis Morin in Paris from 1665 to 1709".*

Because we have extended the data to the period addressed, we would like to keep the title.

*Some other points, indicated below, seem to us to be able to be improved:*

*C2.3-Line 87: it would be preferable to indicate here only the Fontenelle reference. The same passage quoted by Legrand and Le Goff is taken from the original in French*

Thank you for this comment. We will change that.

*C2.4-Line 142 to 164. The authors seem to have completely adopted the method developed by Legrand and Le Goff, 1992 to convert Morin's measurements into °C, a method explained in 10 pages. In this article, the summary which is given in 21 lines does not make it possible to understand the principle, in particular that the 2 periods indicated in line 155 are the years of observation of Morin from 1676 to 1712 and those of the Observatory of Paris from 1816 to 1852 considered to have identical average maximum and minimum temperatures. Are the values   in °C recalculated by the authors identical to those published in volume 2 of the book by Legrand and Le Goff?*

We will provide a more detailed explanation. And yes, the values are identical. (See C1.7 of the first reviewer)

*C2.5-Line 164 and 165: "Rousseau, 2009" and not "Rousseau, 2013"*

Thank you for this comment. We will change that.

*C2.6-Line 166: "grape harvest dates" and not "harvest date"*

Thank you for this comment. We will change that.

*C2.7-Line 172: "monthly" instead of "daily".*

Thank you for this comment. We will change that.

*C2.8-Line 171 and Figure 4: .*

*C2.9-The assessment of a bias in Morin's measurements from 1776 to 1780 [sic, it should be "1676 t0 1680"], which may possibly be taken into account for the calculation of monthly temperatures, is different according to Figure 4 of this article or Figure 1 of the article Rousseau, 2009. It seems that value for 1677 of harvest dates in Beaune used in the article (series of Labbé et al) is not in agreement with the chronology of the temperatures of Morin,*

*which is in phase with the chronology of the Dijon series (series taken from Angot) used in Rousseau, 2009.*

Analogous to Fig. 1 (Rousseau, 2009), we created the same graph (See below). Note that we have excluded September for GST, but Rousseau has included it. This leads to discrepancies in the temperature mean of GST. The red dots show results for the 1681-1713 time period (Linear Fit: Long blue line) and the blue dots show results for the 1676-1680 time period (Linear Fit: Short blue line). The black line shows the linear fit over all data points. We do see higher values for the period of 1676-1680, but far not that pronounced as in Rousseau 2009. Furthermore, the slope of the black fit is well below 1 °C / 10 days. I.e. a distinction between measurement error and climate variability seems to be difficult. As already mentioned in the article: We see the unusually high anomaly but are not sure if it can be homogenized to this extent as in Rousseau 2009. Therefore, we did not homogenize these years, but did point out the high anomalies in temperature. Probably more studies in continental Europe will be needed to differentiate in a more confident way in between inhomogeneity of the data and climate variability. So, we want to keep the raw-data for our analysis, but will discuss this possible anomaly.

[Figure]

*Figure 3: Comparison of grape harvest dates (GHD) with the growing season temperature (GST - Tm). The blue dots show the yearly values from 1676 to 1680. The black line represents the linear regression for all years.*

*C2.10-Line 177 to 179: Writing too technical, difficult to understand. In addition it seems that this smoothing of the data does not allow better readability, the curves somewhat masking the raw data. Wouldn't a data visualization for Figure 5, 8 and 9 similar to that used in Figure 4 be better?*

We checked the visualization as suggested and agree to this comment. Furthermore, the yearly variability can be seen more clearly. This is especially true for the temperature in the DJF season. This means that we can delete the part which is meant to be too technical.

[Figure]

*Figure 4: Directional index (DI) for all seasons*

[Figure]

[Figure]

[Figure]

*Figure 5: Tmax, Tmin and Tmean from 1676 to 1713*

*C2.11-Line 204 and figure A1: For comparison it would be interesting to provide for the Meteoblue data from 1986 to 2015 the 2 Figures similar to those of appendix A1 concerning the Morin data.*

One must differentiate here what is comparable and what is not. Since the ERA5 data are given in full hours, we are limited in this sense and have chosen 15 o'clock for the midday temperature and 19 o'clock for the evening temperature. I.e. with this choice, one will not get precisely the maximum temperature. Thus, for methodological reasons, the absolute values are only comparable to a limited extent. However, two important analogies to Morin's temperature measurement become apparent: (1) There are also years with exceptionally high values (Although the majority is detected in the preliminary version from 1960 to 1978) and (2) the monthly distribution is quite similar to that of Morin. It shows high values in winter months and low values in autumn. A difference can be seen comparing the summer months (June and July), where the calculation of ERA5 data reveals higher values.

[Figure]

*Figure 6: Number of days per year when Tev > Tmi and the monthly distribution of the mean number of days when Tev > Tmi*

Furthermore, the monthly distribution of the remarkable years in terms of high values does not show clear anomalies or similar results when comparing the monthly distribution of those years. See below.

[Figure]

*Figure 7: Monthly distribution of years in Morin's measurements which show a high value in terms of Tev > Tmi*

We will address the differences shortly in our Paper.

*C2.12-Line 221 and figure 3: Is the same difference observed between morning, noon and evening on the Meteoblue data? Morning, noon and evening Meteoblue curves could be shown in Figure 3.*

This comment makes us reconsider this graphic. We were too focused on the mean of the Morin data giving a satisfactory result. However, looking at the results for the midday and evening measurements, one might suspect an inhomogeneity in the measurement series. Looking at the ERA5 results, one sees only small deviations in the result between the different times. Two problems in the methodology arise: (1) Rain or snow was measured by Morin by eye. I.e. if Morin noted snow, then it may have fallen in a specific time period, whereas temperature was read at fixed times. (2) The table of Morin's measurements illustrates another problem: Morin has six possible entry points for precipitation (column 12). At the respective measuring points for the temperature and in between.

Considering Morin's strict daily routine, the entry of the value which is on the line separating day for day should be noted at 2 o'clock in the morning. The next value was noted in line with the first temperature measurement. Thus, (2) can be satisfactorily explained for the morning temperature (1) is not so significant, since the temperature difference between 2 o'clock and 7 o'clock is a smaller one in the winter months. That is, point (2) for the morning has a tendency to minimally decrease the temperature of the 50% snow-rain-threshold. This fact is also reflected in the comparison with the ERA5 data.

However, for the midday and evening temperature these points are more problematic. In case of the midday temperature, it may be that it snowed at lower temperatures (e.g., at 10 a.m.) and then the temperature was measured at the daytime maximum. This also becomes visible in a comparison. For the evening temperature, an attribution is equally more problematic. Here, it seems to us that points (1) and (2) do not matter as much as that we applied Legrand and Le Goff's calibration for the evening temperature. This calibration has been elicited for the maximum temperature and thus, when applied to the evening temperature, will result in higher temperatures at the 50% snow-rain-threshold.

We will address these issues in the paper and elaborate further, as in this response.

[Figure]

*Figure 8: Snowfall frequency of Morin and ERA5 reanalysis data (1960 to 1990)*

*Figure 9: Example of Morin's notes*

*C2.13-Line 235 and figure A2B: the differences noted in the thermal amplitude illustrated in the A2b would deserve comments and undoubtedly a more in-depth study (on a finer scale than the year), which could possibly make it possible to detect more precisely ruptures of the homogeneity of the series, the extreme values   being more sensitive to the local environment of the observation. Figure A2b seems to indicate breaks around 1680, 1690, 1700, 1705 which do not coincide with Morin's changes of domicile. Are deviations of such great amplitude observed in Meteoblue data? Are these differences related to inter-annual variability or to changes in location or others modifications of measure conditions?*

We think that the most remarkable change in terms of DTR appeared in September 1688. Also, the following 2 years are characterized by low values of DTR with especially low values during warm months. We will consider the points raised up in this comment and will look into more detail. E-OBS-data show no year where DTR does not exceed 10 °C from 1950-2019, whereas in the years 1689, 1690 and 1691 DTR stays mostly below 10 °C. So, this may be an inhomogeneity. However, it has to be pointed out that the drop of DTR of about 5°C can be explained because it seems that a cold front appeared. We will indicate this inhomogeneity in the paper and make a proposal for homogenization in the mentioned years.

This means that we want to keep the raw data for the analysis, but in a provided data file we will include a time series of the temperature measurement with the raw data as well as a time series of the temperature measurement with the included homogenizations.

[Figure]

*Figure 10: Diurnal temperature range (DTR) of 1688*

.

*C2.14-Figure 4 – Legend: "Beaune" and not "Dijon"*

Thank you for this comment. We will change that.
* * *
*C2.15-Line 276 and table 2: "the extraordinary positive anomaly" observed between 1676 and 1680, which is not found in the CET temperatures, confirms the hypothesis of a break in the homogeneity of the Morin series in 1680 which could be discussed here*

We discussed the positive anomaly and point out that Rousseau (2009) did a calibration on monthly basis by using the CET temperatures. We are not sure if the calibration can or should be done over the whole year. (See comment 2.9)
* * *
*C2.16-Line 285: The same analysis of cold days for the period 1665 to 1675 seems feasible despite the lesser precision of Morin's small thermometer (distribution of measurements noted f4?) even if it means homogenizing with respect to the following period, due to a different threshold.*

Because of the following points we want to exclude this time period: (1) We have less knowledge of the measurement instrument and (2) some methods we performed to validate the temperature measurements cannot be performed for this time period (Snow-rain-threshold,…). However, we can show cumulative number of values <= f4. Note that Morin used a different scale for 1665-1669 compared to 1670-1675. So, f4 will have different °C values in both time periods (See Rousseau, 2013)

[Figure]

*Figure 11: Impact analysis for DJF*

*C2.17-Line 293: Same comment concerning the hot days (distribution of measurements c4?) of the period 1665-1675*

Same as in C2.16. Note here that for 7 June 1666 to 6 Sept 1666 Morin did not make measurements.

[Figure]

*Figure 12: Impact analysis for JJA*

*C2.18-Line 318: The growth of TCC does not take place over the entire period 1676 to 1709 and therefore the use of a growth rate of TCC from a linear regression established over the entire period is not actually justified. We note very clearly in figure 8 that the 5 curves would be rather decreasing or stationary after 1693*

Thank you for pointing this point out. We will change the graphic. Furthermore, we want to address in the continuous text that the increase may be due to an increase of his notes, because the highest value (4) increases from zero appearances in 1670 to 37 in 1690.

*C2.19-Line 393: Is the only strong deviation of westerly winds observed in the decade 1700-1710 not due to the fact that it is the only decade including a complete phase of positive temperature deviations? The other decades present both rather warm or rather cold temperature phases of multi-decadal fluctuations. It would be interesting to examine whether a division into 4 periods, corresponding to alternately cold and warm phases of multi-decadal*

*fluctuations 1672-1675, 1676-1686, 1687-1701, 1702-1708 (cf Le Roy Ladurie et al., Fluctuations du climat, 2011) would not give clearer differences.*

For comparison, we looked at the indicated time periods in the polar plots. The strength of this illustration is that one can clearly see different distributions (Strong tendency of 1700s to WI, higher frequency of EI in DJF, etc...). However, smaller differences of individual indices are difficult to see. Thus, we will present differences of main wind directions (NI, EI, SI, WI) either in continuous text or in tabular form. However, we would like to keep the presentation of decades because we have focused on decade means in this paper. In particular, the difference in the DJF season between the 1690s and 1700s is salient, where the former has a WI=42% and the latter a WI=58%.

[Figure]

*Figure 13: DI of the seasons and annual with different time periods*

*C2.20-Line 454-455: The complete consideration of temperature data from 1665 to 1675 (winter 1672, summer 1675 remarkable in particular) and from 1710 to 1713 as well as the question of the break in the homogeneity of the measurements suggested by the figure A2b, seems to us to deepen later if the article does not deal with it.*

Thank you for this comment. We will address the last two points as already mentioned in the answers above but will not consider the temperature data from 1665 to 1675 in this paper (See C2.16).

The numbers in the square brackets refer to the first preprint version.

1) [30; 352 ff; 428 ff] Clarify the term Late Maunder Minimum and correct the wrong statement of the citation of Lockwood et al. (2017) (Relates to CC1 and C1.13)
2) [Equ. 4, Equ. 5, Fig. 8-12, multiple lines] Use ERA5-data instead of the Meteoblue-data for comparisons and calculations. [124,…] Deletion of Meteoblue and Inclutson of the ERA5-data. [199 ff] Mention the time triples for the four smallest TI, not just for one. (Relates to C1.2)
3) [Equ. 4, page 10] Change of the indices, and furthermore deletion of the normalization. Thus, each deviation of the comparisons is weighted equally. (Relates to C1.8)
4) [215] Change of the wrong wording. (Relates to C1.10)
5) [Equ. 5, page 12] Deletion of the factor 2. (Relates to C1.11)
6) [349] Being more careful of the difference essential requirement and driving factor. (Relates to C1.13)
7) [367, Fig. 10] Excluding days, when TCC<10% for the calculation of Fig.10. Furthermore, we changed the cloud levels of the ERA5-data to compare the DIs with Morin's data. (Relates to C1.15)
8) [Tab. 1, Tab. 2, Tab. 3; Fig. 3-12; multiple lines] Include the data of 1710–1713. Include the additional years for the calculation or expand the time series for figures, tables, and calculations. Correct naming of the time span. [3, 418] Adding "Institut de France". [134, 139] Update number of digitized data and missing data. (Relates to C2.1)
9) [87] Deletion of redundant references (Relates to C2.3)
10) [153–164] Giving a few more details on the calibration method. (Relates to C3.3 and C1.7)
11) [L164 and 165] Change of Citation (Relates to C2.5)
12) [ 166] Terminology changed into "grape harvest dates" (Relates to C2.6)
13) [172] Wrong wording: "monthly" instead of "daily" (Relates to C2.7)
14) [177–179; Fig. 5, Fig. 8, Fig. 9] Deletion of the section and change of the figure layouts. (Relates to C2.10)
15) [Fig. A1] Adding the mean value of Tev>Tmi from ERA5 (1981–2010) as horizontal line. (Relates to C2.11)
16) [205 ff, Fig. 3] Further explanation and discussion of Fig. 3. (Relates to C2.12 and C1.7)
17) [235 ff, 175 ff] Pointing out an inhomogeneity of the temperature measurements concerning the years 1688 to 1691. Adding a new plot in the appendix. (Relates to C2.13 and C1.3)
18) [Fig. 4] Change "Dijon" to "Beaune"
19) [313 ff, Fig. 9] Deletion of the linear regression and change of the layout of Fig. 9, which includes the time series of each note (0–4).
20)  Added citations concerning the LMM and MM (Relates to CC1 and C1.13)

Additional

1) [7, 316] Change of BE to AE (autumn -> fall)
2) [10] Deletion of one sentence.
3) [425] Adding two sentences, which point out the (small) inhomogeneities of the temperature measurements.
4) Added source to Fig. 2

---

## Author Response (AR3)

Dear editor,

we made the following changes:

[Line 32] Added citations concerning the LMM and MM as mentioned in your response

[Line 33] Added an additional citation, which is not listed in the response:

Luterbacher, J.: The Late Maunder Minimum (1675–1715) — Climax of the 'Little Ice Age' in Europe, 2001

[Fig. 2] Added source to Fig. 2

[Line 529] Added paragraph data availability (DOI of the data file)

[Line 489, Fig A3] Marginal changes

Best regards,

Thomas Pliemon

---

## Author Response (AR4)

Dear editor,

sorry for the incomplete submission. Now all citations are included as follows:

[Line 31 ff] All proposed citations are mentioned/included here.

Best regards,

Thomas Pliemon

---

## Author Response (AR5)

Dear editor,

I corrected some small errors in this final version:

[Fig. 08] Changed the year 1980 to 1981.

[Fig. 11] Changed the year 1709 to 1710.

[Line 530] Added the full doi.

Best regards,

Thomas Pliemon